# Second quantization of many-body dispersion interactions for chemical and biological systems

Matteo Gori[1,2] ✉, Philip Kurian [2] ✉ & Alexandre Tkatchenko [1] ✉

The many-body dispersion (MBD) framework is a successful approach for modeling the long-range electronic correlation energy and optical response of systems with thousands of atoms. Inspired by field theory, here we develop a second-quantized MBD formalism (SQ-MBD) that recasts a system of atomic quantum Drude oscillators in a Fock-space representation. SQ-MBD provides: (*i*) tools for projecting observables (interaction energy, transition multipoles, polarizability tensors) on coarse-grained representations of the atomistic system ranging from single atoms to large structural motifs, (*ii*) a quantum-information framework to analyze correlations and (non)separability among fragments in a given molecular complex, and (*iii*) a path toward the applicability of the MBD framework to molecular complexes with even larger number of atoms. The SQ-MBD approach offers conceptual insights into quantum fluctuations in molecular systems and enables direct coupling of collective plasmon-like MBD degrees of freedom with arbitrary environments, providing a tractable computational framework to treat dispersion interactions and polarization response in intricate systems.

Noncovalent interactions[1–4] play a key role in determining physico-chemical properties, given that they influence the structure[5], stability[6,7], dynamics[8–10], and electric[11] and optical[12] responses in a wide range of molecules and materials[13–15]. In particular, van der Waals (vdW) dispersion interactions and long-range electron correlation energy must be treated with quantitative many-body methods[16–24]. Different methods[25–29] have been proposed to include dispersion interactions in the form of non-local vdW density functionals. The many-body dispersion (MBD) framework[30,31] has been firmly established as an efficient and accurate approach. In MBD, the electronic response properties of each atom are represented by a quantum Drude oscillator (QDO)[32]. The long-range correlations among the electronic fluctuations emerge from the dipolar coupling between the QDOs. The MBD method can be now routinely applied to systems with up to $N \sim 10^4$ atoms[9], a size limitation owing to the $N^3$ computational scaling of MBD. Furthermore, MBD effects have been shown to extrapolate to mesoscale processes[33,34], including solvation and folding of proteins[9,35]

and the delamination of graphene from surfaces[36], demonstrating the interplay between MBD modes and collective nuclear vibrations[34,37]. These findings suggest that MBD interactions contribute to coopera-tive effects between electronic and nuclear degrees of freedom in complex chemical and biophysical systems. These effects include non-local allosteric pathways in enzymes from coordinated electronic fluctuations[38–40] and the emergence of giant electric-dipole oscillations in biomolecules that mediate long-range intermolecular interactions[41,42]. Pursuing the study of MBD effects in realistic systems in complex environments requires simulations with millions of atoms, which are infeasible at the moment even with stochastic implementations[35,43]. The development of a coarse-grained MBD model would be a compelling strategy to provide a conceptual and computational leap to extend the applicability of MBD to systems with a large number of atoms. With this goal in mind, we propose here a second quantization formulation of the MBD model (SQ-MBD) that considerably simplifies the calculation of the fragment contributions

[1]Department of Physics and Materials Science, University of Luxembourg, L-1511 Luxembourg City, Luxembourg. [2]Quantum Biology Laboratory, Howard University, Washington, DC 20060, USA. ✉e-mail: matteo.gori@uni.lu; pkurian@howard.edu; alexandre.tkatchenko@uni.lu

to observables stemming from collective MBD modes, enhances physical intuition on how MBD effects operate to connect different length scales in atomistic systems and establishes a strong connection between the MBD method and quantum information theory. For instance, the fragment contribution to the total MBD energy could be used as reference data for machine-learned force fields[44–46], while coarse-grained fragment contributions to transition dipole elements of excited MBD states could yield effective models to predict collective optical response in biomolecular complexes[47–50] and J-aggregates[51,52]. In addition, quantum information observables calculated from the SQ-MBD Hamiltonian indicate that specific structural elements (e.g., certain amino acids in a protein) play a key role in the structural stability of folded states and might contribute to improve our understanding of the role of mutations and intra- and intermolecular allosteric pathways in biomolecules.

## Results

### The MBD method in a nutshell

We investigate non-covalent dispersion interactions, specifically interatomic interactions resulting from correlations among quantum fluctuations of electronic charge density. The adiabatic connection fluctuation-dissipation (ACFD) theorem provides an exact formula expressing the correlation-dissipation energy as a function of the electronic density response of non-interacting and mutually interacting electrons[53,54]. For systems with bound electrons, the ACFD theorem can be reformulated using the non-local polarizability tensor, denoted as $\mathcal{A}^{(\lambda)}$, as follows:

$$E_c = \frac{1}{2\pi} \int_0^1 d\lambda \int_0^{+\infty} \int_{\mathbb{R}^3} d\boldsymbol{r} \int_{\mathbb{R}^3} d\boldsymbol{r}' \ \mathrm{Tr}\left[ \left( \mathcal{A}^{(\lambda)}(\boldsymbol{r},\boldsymbol{r}',i\omega) + \right. \right.$$
$$\left. \left. -\mathcal{A}^{(0)}(\boldsymbol{r},\boldsymbol{r}',i\omega) \right) \mathbb{T}(\boldsymbol{r},\boldsymbol{r}') \right] d\omega \quad (1)$$

where the non-local polarizability $\mathcal{A}^{(\lambda)} = \mathcal{A}^{(0)} - \mathcal{A}^{(0)}(\lambda\mathbb{T} + \nabla_{\boldsymbol{r}} \otimes \nabla_{\boldsymbol{r}} f_{xc}^{\lambda})\mathcal{A}^{(\lambda)}$ is the solution of the Schwinger–Dyson equation with exchange-correlation functional $f_{xc}^{\lambda}$ and the dipole–dipole potential $\mathbb{T}$ (see the "Methods" section for further details). Adopting the random phase approximations (RPA), i.e. $f_{xc}^{\lambda} = 0$, the dispersion energy for a system of $N$ atoms with fixed nuclear positions $\{\boldsymbol{R}_A\}_{A=1}^N$ can be accurately and efficiently estimated by representing the valence electronic response properties of atom $A$ by a 3D isotropic QDO parametrized by its angular frequency $\omega_A$, mass $m_A$, and electric charge $Z_A e$. The approximation of isotropic QDOs is widely accepted in the framework of the range-separated self-consistent screening MBD@rsSCS method that we have adopted in our work. The hypothesis of an isotropic atomic response yields the assumption that the step function included in the potential for the range separation depends only on the interatomic distance. Including anisotropic QDO polarizability would yield an anisotropic step function for the range separation, i.e., depending on both the direction and modulus of the interatomic distance vector. The solution of the technical aspects related to a consistent extension of the range-separation method to such a case is beyond the scope of the present work but has been suggested in ref. 55. Within this framework, the polarizability of non-mutually interacting valence electrons can be effectively represented by a localized field of QDOs, i.e. $\mathcal{A}^{(0)}(\boldsymbol{r},\boldsymbol{r}',i\omega) = \mathcal{A}_A^{(0)}(i\omega)\delta^3(\boldsymbol{r}-\boldsymbol{R}_A)\delta^3(\boldsymbol{r}'-\boldsymbol{R}_B)\delta_{AB}$ where $\mathcal{A}_A^{(0)}(i\omega) = \mathcal{A}_A^{(0)}(1+\omega^2/\omega_A^2)^{-1}$ is the QDO polarizability. It has been proved that for a set of QDOs, the correlation energy can be calculated from an equivalent Hamiltonian formulation[56].

The MBD Hamiltonian takes the form

$$\hat{H}_{MBD} = \frac{1}{2} \sum_{A=1}^N \left[ \| \hat{\boldsymbol{p}}_A \|^2 + \sum_{B \neq A} \hat{\boldsymbol{q}}_A \mathbb{V}_{AB} \hat{\boldsymbol{q}}_B \right] \quad (2)$$

where $\hat{\boldsymbol{q}}_A = \sqrt{m_A}\hat{\boldsymbol{r}}_A = (\hat{q}_{A_{x_1}}, \hat{q}_{A_{x_2}}, \hat{q}_{A_{x_3}})$ is the (mass-weighted) displacement operator of the QDOs associated with atom $A$, and the momentum operator $\hat{\boldsymbol{p}}_A$ is the associated canonical conjugate variable.

The interaction potential is described by the $3 \times 3$ matrices $\mathbb{V}_{AB} = \omega_A\omega_B[\mathbb{I}\delta_{AB} + \sqrt{\mathcal{A}_A^{(0)}\mathcal{A}_B^{(0)}}\ \mathbb{T}_{AB}(\boldsymbol{R}_{AB})]$ where $\mathbb{I}$ and $\mathbb{T}_{AB}(\boldsymbol{R}_{AB})$ are, respectively, the $3 \times 3$ identity matrix and the dipole–dipole coupling between QDOs of atom $A$ and atom $B$ ($\boldsymbol{R}_{AB} = \boldsymbol{R}_B - \boldsymbol{R}_A$). It should be stressed that the dipole–dipole approximation has been used in writing the MBD Hamiltonian in Eq. (2). An active area of research concerns methods to accurately and efficiently estimate the effects on corrections to the MBD energy of higher-order terms from the multipolar expansion of coupling between quantum electronic density fluctuations[57–59]. In this work, we will apply the MBD method to study highly polarizable supramolecular systems and a small protein representative of larger biomolecular systems. At the first order in perturbation theory, it has been shown that multipolar effects are negligible for small biological dimers[57]. As multipolar effects tend to be short-ranged, it can be safely assumed that they can be neglected in larger biomolecular complexes[57]. In supramolecular 3D complexes, it has been demonstrated that the multipolar terms make a significant contribution to the binding energy, but their magnitude remains below 15–20% of the contribution due to dipole–dipole interactions[57]. Such a level of accuracy is sufficient for this work, which focuses on the insights that a second-quantized reformulation of the MBD model can afford.

The Hamiltonian in Eq. (2) is quadratic in the QDO variables so that it can be reduced to the normal form $\hat{H}_{MBD} = (1/2)\left[\sum_{k=1}^{3N} \hat{\tilde{p}}_k^2 + \tilde{\omega}_k^2\hat{\tilde{q}}_k^2\right]$ where $\hat{\tilde{q}}_k, \hat{\tilde{p}}_k, \tilde{\omega}_k$ are, respectively, the displacement, the momentum, and the angular frequency of the $k$th normal mode. We assume in what follows that $0 < \tilde{\omega}_k \leq \tilde{\omega}_{k+1}$ for all $k = 1, \dots, 3N$. The canonical transformation from the atomic-based operators $\{\hat{\boldsymbol{q}}_A, \hat{\boldsymbol{p}}_A\}_A$ to the $3N$ MBD normal-mode variables $\{\hat{\tilde{q}}_k, \hat{\tilde{p}}_k\}_k$ is determined by the orthogonal matrix $\mathcal{O}$ that diagonalizes the potential matrix in the MBD Hamiltonian (see Fig. 1a). In this framework, the MBD energy is given by the energy difference between the interacting QDOs and the QDOs at infinite separation, $E_{MBD} = (\hbar/2)\left[\sum_{k=1}^{3N} \tilde{\omega}_k - 3\sum_{A=1}^N \omega_A\right]$. Although the coupling among QDOs is pairwise in the MBD Hamiltonian in Eq. (2), the MBD eigenfrequencies depend on all the atomic coordinates, $\tilde{\omega}_k^2 = \tilde{\omega}_k^2(\boldsymbol{R}_1, \dots, \boldsymbol{R}_N)$. Consequently, the MBD energy exhibits a many-body nature, being contingent upon the overall system configuration and not reducible to a summation of individual pairwise contributions among the atoms, i.e., $E_{MBD} = E_{MBD}(\boldsymbol{R}_1, \dots \boldsymbol{R}_N) \neq \sum_{A,B} f_2(\boldsymbol{R}_A, \boldsymbol{R}_B)$. The results presented in the following sections have been obtained using a version of the MBD Hamiltonian in Eq. (2) modified according to the widely adopted range-separated self-consistent screening MBD (MBD@rsSCS) method (see the "Methods" section for further details). Such a modification consists essentially of a reparametrization of the atomic QDOs and the addition of a damping factor to the dipole–dipole interaction matrix. These adjustments serve the dual purpose of avoiding divergences from short-range interactions and correctly reproducing the screening effects induced by the local atomic environment.

### Second quantization formulation of the MBD method (SQ-MBD)

Although the orthogonal matrix $\mathcal{O}$ and the set of eigenenergies $\hbar\tilde{\omega}_k$ fully characterize the MBD ground state, such a first quantization approach is not optimal to analyze how collective quasi-plasmonic modes emerge from the correlations among atomic QDOs excitations. From a computational perspective, this yields a cumbersome calculation of matrix elements of localized observables, i.e. depending on atomic QDOs degrees of freedom, in the MBD Hamiltonian eigenstates. A possible strategy to highlight the connection between local

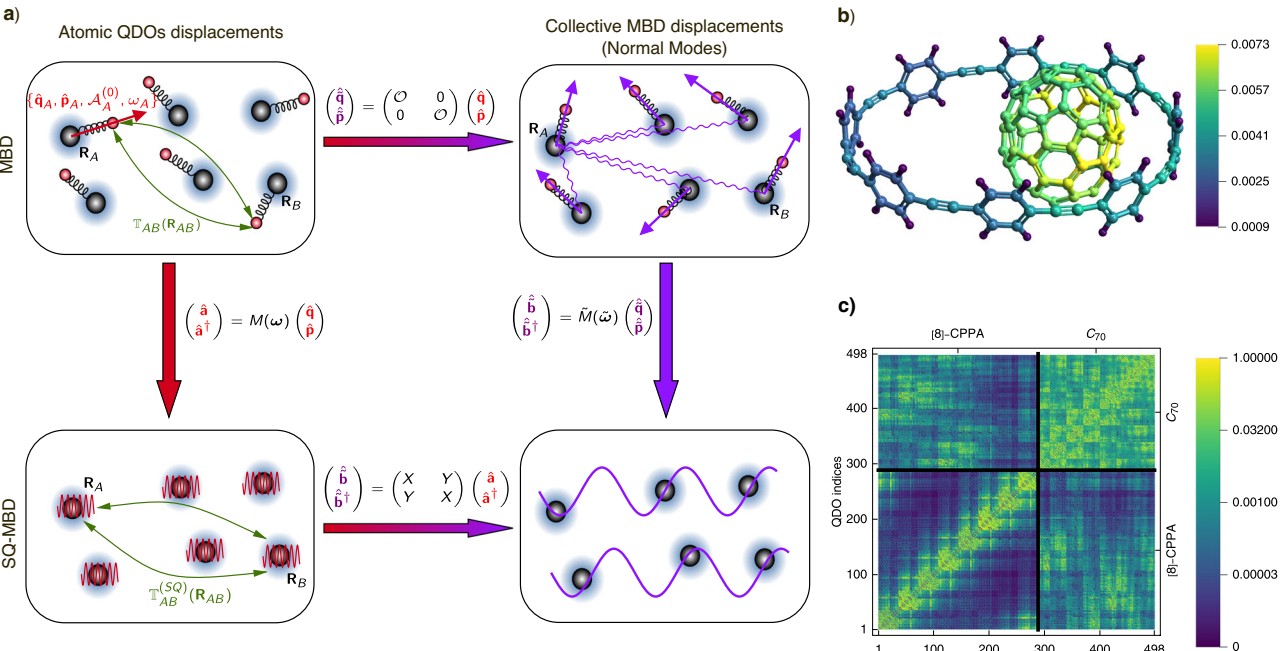

**Fig. 1 | Theory and practice of the second quantization formulation of the many-body dispersion (SQ-MBD) method.** Panel **a** illustrates a commutative diagram outlining the connection between the original many-body dispersion (MBD) framework and its second-quantized formalism (SQ-MBD). Within the MBD method, the electric response characteristics of atom A positioned at $R_A$ can effectively be described using a quantum Drude oscillator (QDO). Many-body dispersion interactions originate from correlation among QDOs due to dipole–dipole interactions represented by the tensor $\mathbb{T}_{AB}(R_{AB})$. In first-quantization formalism, the normal modes displacements $\hat{\tilde{q}}$ and momenta $\hat{\tilde{p}}$ of the MBD Hamiltonian can be expressed in terms of the atomic QDOs degrees mass-weighted displacement operators $\hat{q}$ and their conjugate momenta $\hat{p}$, through an orthogonal transformation represented by the matrix $\mathcal{O}$. In the SQ-MBD framework, the creation/annihilation operators $\hat{\tilde{b}}, \hat{\tilde{b}}^\dagger$ of the MBD collective modes can be expressed in terms of the creation/annihilation operators $\hat{a}, \hat{a}^\dagger$ of the atomic QDOs. $M$ and $\tilde{M}$ matrices denote transformation matrices between first- and second- quantization representations for the atomic QDOs and the collective MBD modes, respectively. The SQ-MBD the dipole–dipole coupling operator can be written as $\mathbb{T}_{AB}^{(SQ)}(R_{AB})$. The matrices $X$ and $Y$ define the Bogoliubov transformation between the creation/annihilation operators of the atomic QDOs and the collective MBD modes. Panel **b** shows the mean excitation numbers of atomic QDOs in the MBD ground state for the supramolecular complex of $C_{70}$ fullerene surrounded by a cycloparaphenyl ring composed of 8 units ([8]-CPPA). Panel **c** shows the normalized covariance matrix of excitation numbers for interacting atomic quantum Drude oscillators (QDOs), decomposed into single Cartesian components. The index $3(A-1) + i$ is assigned to the QDO associated with the displacement of the Drude particle on the $A$th atom along the $i$th Cartesian direction. Source data are provided as Source Data files.

and global fluctuations of the QDO charge density consists of representing the quanta of collective MBD normal modes (the MBD plasmon-like quasiparticles) as a linear combination of the non-interacting atomic QDOs Hamiltonian eigenstates.

The second quantization framework (SQ-MBD) developed here overcomes these issues, providing a description of the degrees of freedom of the atomic QDOs and of the MBD collective plasmonic modes in terms of the algebra of ladder operators for isolated QDOs $\{\hat{a}_{A_{x_i}}, \hat{a}^\dagger_{A_{x_i}}\}_{A,i}$ with the associated basis set $|\boldsymbol{n}\rangle = \bigotimes_{A,i} |n_{A_{x_i}}\rangle$, and the algebra of ladder operators for coupled QDOs $\{\hat{b}_k, \hat{b}^\dagger_k\}_k$ with the basis set $|\bar{\boldsymbol{n}}\rangle = \bigotimes_k |\bar{n}_k\rangle$. Owing to the linear transformation $M(\boldsymbol{\omega})(\tilde{M}(\tilde{\boldsymbol{\omega}}))$ from atomic QDO (normal mode) displacements/momenta operators to the corresponding set of ladder operators for the collective MBD modes, it is possible to construct the commutative diagram reported in Fig. 1a, with an explicit expression for the mapping between the two algebras of creation/annihilation operators, in terms of the orthogonal matrix $\mathcal{O}$ and two sets of eigenfrequencies $\{\omega_A\}_A$ and $\{\tilde{\omega}_k\}_k$, given by

$$\begin{pmatrix} \hat{\boldsymbol{b}} \\ \hat{\boldsymbol{b}}^\dagger \end{pmatrix} = \begin{pmatrix} X & Y \\ Y & X \end{pmatrix} \begin{pmatrix} \hat{\boldsymbol{a}} \\ \hat{\boldsymbol{a}}^\dagger \end{pmatrix}, \tag{3}$$

where $X(\mathcal{O},\boldsymbol{\omega},\tilde{\boldsymbol{\omega}}), Y(\mathcal{O},\boldsymbol{\omega},\tilde{\boldsymbol{\omega}})$ are $3N \times 3N$ real matrices (see section I of the Supplementary Discussion for further details). The linear map in Eq. (3) is a multimodal Bogoliubov transformation[60,61], preserving the canonical commutation relations of the ladder operator algebra.

Bogoliubov transformations in finite quantum systems admit a unitary representation[61] $\hat{S}^{-1} = \hat{S}^\dagger$ such that $\hat{b}_k^{(\dagger)} = \hat{S}\hat{a}_{A_{x_i}}^{(\dagger)}\hat{S}^{-1}$ for $k = 3(A-1) + i$, connecting the ground state $|\boldsymbol{0}\rangle = \bigotimes_{A,i} |0_{A_{x_i}}\rangle$ of the uncoupled atomic QDO system with the collective MBD ground state

$$\left| \tilde{\boldsymbol{0}} \right\rangle = \hat{S}|\boldsymbol{0}\rangle = \frac{e^{\frac{1}{2}\sum_{A,B=1}^{N}\sum_{i,j=1}^{3} \hat{a}^\dagger_{A_{x_i}} \Theta_{A_{x_i}B_{x_j}} \hat{a}^\dagger_{B_{x_j}}}}{\det^{1/4}(XX^{\mathsf{T}})} |\boldsymbol{0}\rangle, \tag{4}$$

where $\Theta = X^{-1}Y$ is a $3N \times 3N$ symmetric real matrix. Equations (3) and (4) represent the information encoded in the MBD ground state in terms of the excited states of the atomic QDOs. The SQ-MBD Hamiltonian thus reads

$$\hat{H}_{SQ-MBD} = \sum_{k=1}^{3N} \hbar \tilde{\omega}_k \left( \hat{b}^\dagger_k \hat{b}_k + \frac{1}{2} \right). \tag{5}$$

In what follows, we present applications of the SQ-MBD framework to the analysis of the MBD ground state properties in supramolecular and biological systems. For all the results that we present, the atomic QDOs frequencies $\boldsymbol{\omega}$, the MBD eigenfrequencies $\tilde{\boldsymbol{\omega}}$ and the orthogonal matrix of MBD eigenvectors $\mathcal{O}$ have all been derived using the current state-of-the-art MBD@rsSCS approach (see the "Methods" section for further details). This ensures the consistency of our total SQ-MBD energies with the MBD@rsSCS method.

## Visualization of the MBD ground state

We first analyze the excitation numbers of atomic QDOs in the many-body state defined as $\langle \hat{\mathcal{N}}_A \rangle_{\tilde{\boldsymbol{O}}} = \sum_{i=1}^{3} \langle \tilde{\boldsymbol{O}} | \hat{a}^{\dagger}_{A_{x_i}} \hat{a}_{A_{x_i}} | \tilde{\boldsymbol{O}} \rangle$ as well as the pairwise correlations between excitation numbers of atomic QDOs $\mathrm{Cov}_{\tilde{\boldsymbol{O}}}(\hat{\mathcal{N}}_{A_{x_i}}, \hat{\mathcal{N}}_{B_{x_j}}) = \langle \hat{\mathcal{N}}_{A_{x_i}} \hat{\mathcal{N}}_{B_{x_j}} \rangle_{\tilde{\boldsymbol{O}}} - \langle \hat{\mathcal{N}}_{A_{x_i}} \rangle_{\tilde{\boldsymbol{O}}} \langle \hat{\mathcal{N}}_{B_{x_j}} \rangle_{\tilde{\boldsymbol{O}}}$, which are reported in Fig. 1b and c for a complex of $C_{70}$ fullerene surrounded by an [8]-CPPA molecular ring. The $C_{70}$-CPPA system constitutes a benchmark for the calculation of the dispersion energy[57], given that it is essentially homonuclear and highly polarizable. The correlations have been normalized as follows $\overline{\mathrm{Cov}}_{\tilde{\boldsymbol{O}}}(\hat{\mathcal{N}}_{A_{x_i}}, \hat{\mathcal{N}}_{B_{x_j}}) = \mathrm{Cov}_{\tilde{\boldsymbol{O}}}(\hat{\mathcal{N}}_{A_{x_i}}, \hat{\mathcal{N}}_{B_{x_j}}) / \sqrt{\langle \hat{\mathcal{N}}_{A_{x_i}} \rangle_{\tilde{\boldsymbol{O}}} \langle \hat{\mathcal{N}}_{B_{x_j}} \rangle_{\tilde{\boldsymbol{O}}}}$. Such a normalization gives $\overline{\mathrm{Cov}}_{\tilde{\boldsymbol{O}}}(\hat{\mathcal{N}}_{A_{x_i}}, \hat{\mathcal{N}}_{A_{x_i}}) = 1$ in the case of a Poissonian distribution for excitations in a single QDO. For all the atomic QDOs in the complex, the values of the atomic mean excitation number are below $10^{-2}$, suggesting that the dipolar interactions in the MBD ground state act as a perturbation on the uncoupled atomic QDO system. Such an observation is confirmed by the strength of normalized correlations between QDOs not exceeding 0.2. The QDOs on the fullerene have higher mean excitation numbers, and their mutual correlations are stronger compared with the atomic QDOs associated with the carbon atoms in the CPPA molecule. This suggests that collective effects are stronger in a compact quasi-spherical homonuclear fullerene. The asymmetry of the fullerene position with respect to the CPPA ring manifests itself in an enhancement of the excitation of the $C_{70}$ QDOs located closer to the phenyl rings. This can be interpreted as a polarization effect on the fullerene due to the CPPA acting as an external environment. Similar conclusions are reached when the fullerene is considered as an external environment acting on the CPPA. Excitation number analysis in the SQ-MBD framework can be easily extended to more complex systems, providing a useful tool to investigate the effect of a general environment on coupled QDOs, which can be used for the development of effective models of MBD interactions in open systems.

## Local contribution to MBD interaction energy in biomolecular complexes

The SQ-MBD framework also considerably simplifies the calculation of operator expectation values between fragments in the collective MBD state. Let us consider a partition $\{\mathcal{F}_{\alpha}\}_{\alpha=1}^{N_{\mathrm{frag}}}$ of the whole system, $\mathcal{S} = \cup_{\alpha} \mathcal{F}_{\alpha}$, each fragment being specified by a set of atomic QDOs, $\mathcal{F}_{\alpha} = \{A_1, \ldots, A_{N_{\alpha}}\}$. For such a partition, it is possible to define the single- and pair-fragment contributions to the total MBD energy of the system $E$,

$$E_{\mathrm{MBD}} = \sum_{\alpha,\beta=1}^{N_{\mathrm{frag}}} (E_{\mathrm{MBD}})_{\alpha\beta} = \sum_{\alpha=1}^{N_{\mathrm{frag}}} \left( E^{(\mathrm{frag})}_{\mathrm{MBD}} \right)_{\alpha} \qquad (6)$$

where $(E_{\mathrm{MBD}})_{\alpha\alpha} = (U_{\mathrm{MBD}})_{\alpha} = \langle \tilde{\boldsymbol{O}} | \hat{H}_{\mathrm{MBD}} |_{\mathcal{F}_{\alpha}} | \tilde{\boldsymbol{O}} \rangle - (\hbar/2) \sum_{A \in \alpha} \omega_A$ is the internal MBD energy of the $\alpha$th fragment, $(E_{\mathrm{MBD}})_{\alpha\beta} = (V_{\mathrm{MBD}})_{\alpha\beta} = 1/2 \times \langle \tilde{\boldsymbol{O}} | \hat{H}_{\mathrm{MBD}} |_{\mathcal{F}_{\alpha} \cup \mathcal{F}_{\beta}} | \tilde{\boldsymbol{O}} \rangle$ is the mutual MBD interaction energy between the $\alpha$th and $\beta$th fragments with $\alpha \neq \beta$, and $(E^{(\mathrm{frag})}_{\mathrm{MBD}})_{\alpha} = \sum_{\beta=1}^{N_{\mathrm{frag}}} (E_{\mathrm{MBD}})_{\alpha\beta}$ is the total contribution of the $\alpha$th fragment to the total MBD energy. We stress that such quantification of the fragment contribution to the collective MBD energy is not unique since there are multiple ways to partition the pair-fragment contributions. However, the proposed fragment-based projection scheme can be straightforwardly applied to large molecular systems with arbitrary levels of coarse-graining[62–64]. As a case study, here we consider crambin (see Fig. 2), a protein with 46 amino acid residues exhibiting essentially all relevant secondary-

structure motifs, and that has been extensively used as a model for crystallography, NMR technique development, and folding studies[65]. The energy scale of single-residue contributions to the MBD energy is $|(E^{(\mathrm{frag})}_{\mathrm{MBD}})_{\alpha}| \sim 0.1 - 1 \, \mathrm{eV}$, while for larger secondary-structure elements $|(E^{(\mathrm{frag})}_{\mathrm{MBD}})_{\alpha}| \sim 0.5 - 9 \, \mathrm{eV}$–as strong as covalent bonds. This reinforces the relevance of dispersion interactions and their interplay with covalent bonding in driving the dynamics of biomolecular systems. Interestingly, there are fragments exhibiting a *positive* internal MBD energy both in the case of residues (see Fig. 2b) and of secondary structures (see Fig. 2e). In particular, the residues with a positive internal MBD energy represent a large majority (33 of 46), while the only secondary structure motif with positive $(U_{\mathrm{MBD}})_{\alpha}$ is the link2 structure. This effect can be interpreted as the screening of the intrafragment MBD interactions due to the presence of the external environment, leading to a blueshift of atomic QDO frequencies not compensated by the negative energy contribution due to the mutual dipolar interactions between QDOs inside the fragment. However, the total single-fragment contribution to the MBD energy for all the fragments in the biomolecule is negative for the considered coarse-graining schemes (see Fig. 2a and d). Particularly suggestive is the case of the phenylalanine residue (PHE13) located at the center of the longest alpha-helix: it is a fragment with a high positive internal MBD energy $((U_{\mathrm{MBD}})_{\mathrm{PHE13}} \sim 0.13 \, \mathrm{eV})$, and it is also the residue with the largest negative single-fragment energy contribution $((E^{(\mathrm{frag})}_{\mathrm{MBD}})_{\mathrm{PHE13}} \sim -1.1 \, \mathrm{eV})$. This can be interpreted as a fingerprint of the strong coupling of the fragment with the external environment: the correlations among atomic QDOs inside the fragment are disturbed in favor of establishing a stronger correlation with the rest of the protein. On the other hand, the difference between single-fragment contributions to MBD and the internal MBD energy for the longest helix in the complex is $\sim -2.2 \, \mathrm{eV}$ compared with $(U_{\mathrm{MBD}})_{\mathrm{helix1}} \sim -6.3 \, \mathrm{eV}$. This means that the QDOs inside the alpha-helix are strongly correlated and interacting, constituting a fragment weakly coupled to the rest of the protein. This analysis suggests a possible strategy to develop a coarse-grained model of MBD interactions similar to existing quantum embedding methods[66–69]: identifying the fragments with stronger internal correlations and interactions among atomic QDOs, solving the coupled QDOs inside these fragments and treating the weaker interactions with the rest of the system in a perturbative way, in analogy with the orbital hybridization description of covalent interactions. The $N_{\mathrm{frag}} \times N_{\mathrm{frag}}$ matrix $V_{\mathrm{MBD}}$ and the $N_{\mathrm{frag}}$-dimensional vectors $U_{\mathrm{MBD}}, E^{(\mathrm{frag})}$ provide low-rank representations of MBD interactions and could be used to develop these coarse-grained models, which can serve as inputs to machine-learned force fields[44–46]. Finally, it is worth to notice the directional and selective character of the coarse-grained MBD interactions in crambin. On one side, in fact, certain residues exhibit significantly stronger inter-fragment interactions compared to others, thereby exhibiting a selectivity condition. On the other side, the interaction energy among residues is not isotropic, further indicating the presence of directionality.

## "Bonding" and "anti-bonding" contributions from SQ-MBD analysis of supramolecular complexes

The methods presented in the previous subsection provide a decomposition of the interaction energy contained in the fully coupled MBD state of the whole system without the need for arbitrary projections into MBD eigenstates of isolated fragments, as it is usually done[9,70]. Let us consider, for instance, the $(\hat{V}_{\mathrm{MBD}})_{\mathcal{AB}}$ operator describing the interaction between two monomers $\mathcal{A}$ and $\mathcal{B}$ in a supramolecular dimer. Thanks to the SQ-MBD expression of $(V_{\mathrm{MBD}})_{\mathcal{AB}} = \langle \tilde{\boldsymbol{O}} | (\hat{V}_{\mathrm{MBD}})_{\mathcal{AB}} | \tilde{\boldsymbol{O}} \rangle$, it is possible to evaluate the contribution of each MBD mode to the inter-fragment MBD energy $(V_{\mathrm{MBD}})_{\mathcal{AB}} = \sum_{k=1}^{3N} (V_{\mathrm{MBD}})_{\mathcal{AB},k}$ (see section III of the Supplementary Discussion). The significance of this analysis lies in identifying the key collective MBD modes responsible for the

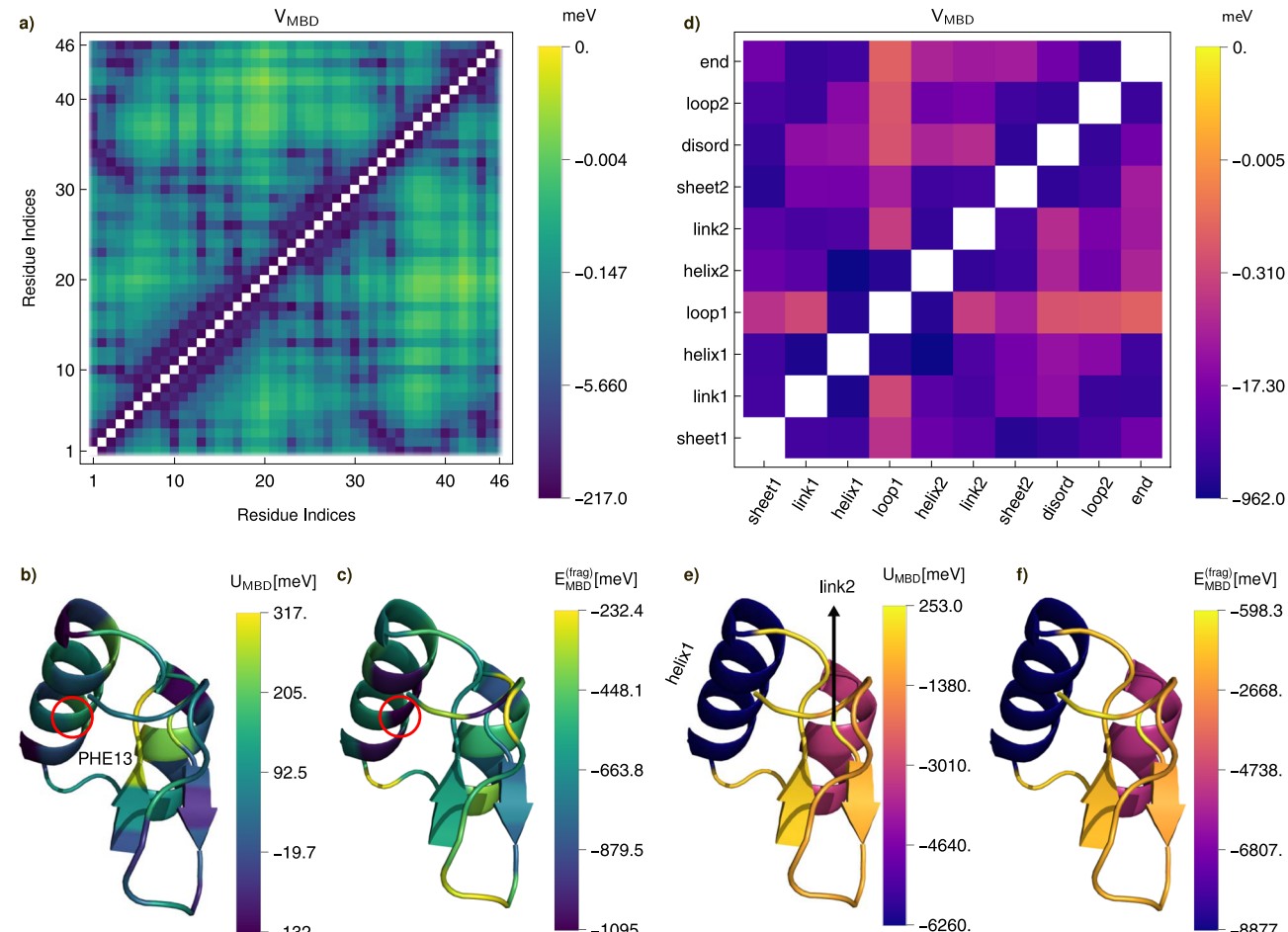

**Fig. 2 | Contributions to the many-body dispersion (MBD) energy for the crambin protein [PDB ID: 2FD7] from two different coarse-grained partitions (fragments) of atomic quantum Drude oscillators (QDOs).** In the left panels, **a** the interfragment contribution ($V_{MBD}$) to the mutual interaction energy between fragments in MBD, **b** the internal MBD energy of each fragment ($U_{MBD}$), and **c** the total fragment contribution to the MBD energy ($E_{MBD}^{(frag)}$) are presented for a partitions of the considered crambin structure into residues. The same quantities are presented in panels **d**–**f** for the specified secondary-structure fragments. All energy values are expressed in meV. Source data are provided as a Source Data file.

interfragment MBD energy within a specific geometry, eliminating the need to compute and project into MBD modes for isolated systems. To illustrate the insights that can be derived from such an analysis, we consider two supramolecular complexes: the C70@[8]-CPPA complex considered in Fig. 1, and a "tweezer" complex dominated by dispersion interactions taken from the S12L database, already studied in ref. 70.

It is important to emphasize that our proposed SQ-MBD analysis for assessing the MBD eigenmodes contribution to interfragment MBD energy only involves computing the MBD eigenmodes for a single geometry. This approach differs qualitatively from other MBD analyses on supramolecular complexes, such as the one presented in ref. 70. The latter focuses on the most significant MBD eigenmode contributions to the MBD-binding energy, which is a distinct observable compared to the interfragment MBD potential energy, and requires comparing a given configuration with another one involving fragments at an infinite distance.

Figure 3 presents the SQ-MBD analysis of the MBD eigenmode contribution to the mutual MBD energy between monomers, $(V_{MBD})_{\mathcal{AB},k} = (V_{MBD})_{\mathcal{AB}}(\tilde{\omega}_k)$ for both considered structures. The first observation is that the total mutual interaction energy among the fragments is negative $\sum_{k=1}^{3N}(V_{MBD})_{\mathcal{AB},k} < 0$ for both structures. For instance, in the "tweezer" complex ($\sim 10^2$ atoms), the total MBD-binding energy is $E_{bind,MBD} \approx -0.818$ eV, while the interfragment interaction energy in the MBD ground state is $(V_{MBD})_{\mathcal{AB}} \approx -1.67$ eV. This difference arises because in SQ-MBD we compute the expectation value of the

interfragment interaction energy evaluated on the fully coupled MBD state. The second (deeper and highly nontrivial) insight is that the MBD eigenmodes can be distinguished as either "bonding" MBD modes with a negative contribution to the interaction energy ($(V_{MBD})_{\mathcal{AB},k} < 0$) or "antibondin" MBD modes with a positive contribution ($(V_{MBD})_{\mathcal{AB},k} > 0$). Furthermore, in both systems, we observe a clear pattern in the distribution of contributions to the interfragment potential energy $(V_{MBD})_{\mathcal{AB},k}$ as a function of the eigenfrequencies $\tilde{\omega}_k$. The most significant bonding MBD modes tend to occur at lower eigenfrequencies, while the most significant antibonding MBD modes are found at higher MBD eigenfrequencies. In the intermediate frequency range, there is a high density of modes that contribute almost zero to the interaction energy (see Fig. 3). It is interesting to notice that there is no evident asymmetry between the bonding and antibonding MBD modes. Moreover, the SQ-MBD formalism not only allows identification of the bonding/antibonding character of the noncovalent MBD modes, but it also allows calculation and visualization of the contribution of each pair of atomic QDOs to $(V_{MBD})_{\mathcal{AB},k}$ for a given MBD mode, as reported in the insets of Fig. 3. These help to identify which constituents contribute most significantly to the quantum collective fluctuations and, consequently, the most relevant electronic degrees of freedom that participate in defining the dimer geometry.

These insights on MBD eigenmodes and their contributions to the binding energy could be used to develop advanced coarse-graining procedures, per the analysis shown in Fig. 2. Hence, SQ-MBD offers a

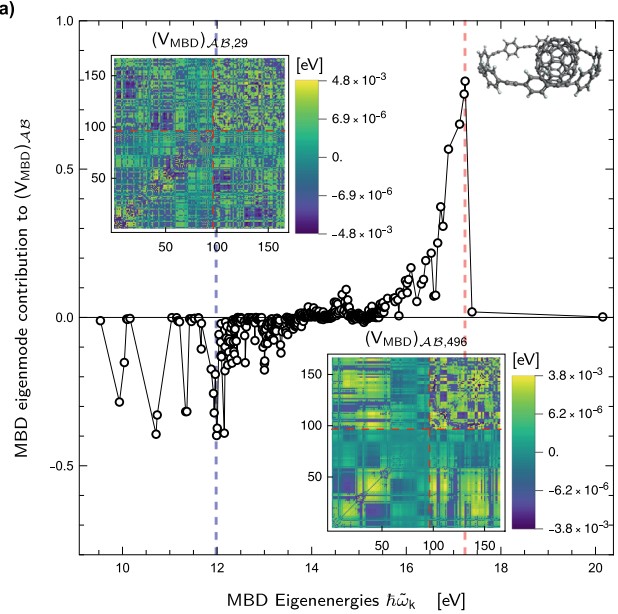

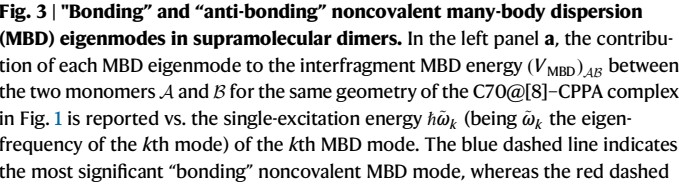

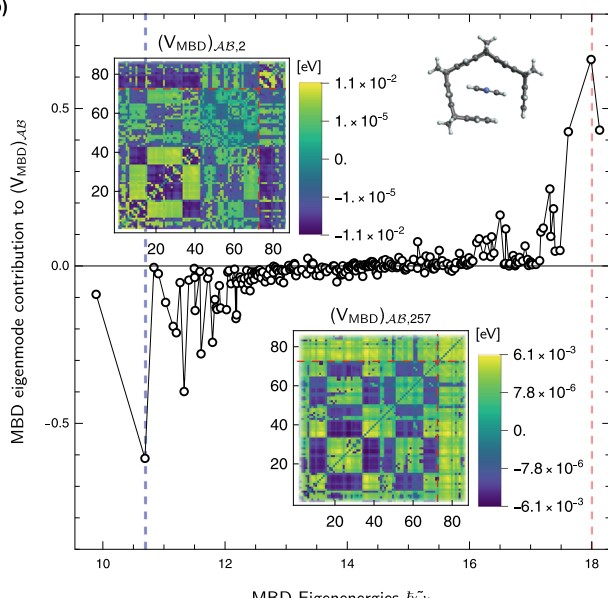

**Fig. 3 | "Bonding" and "anti-bonding" noncovalent many-body dispersion (MBD) eigenmodes in supramolecular dimers.** In the left panel **a**, the contribution of each MBD eigenmode to the interfragment MBD energy $(V_{MBD})_{\mathcal{AB}}$ between the two monomers $\mathcal{A}$ and $\mathcal{B}$ for the same geometry of the C70@[8]–CPPA complex in Fig. 1 is reported vs. the single-excitation energy $\hbar\omega_k$ (being $\tilde\omega_k$ the eigenfrequency of the $k$th mode) of the $k$th MBD mode. The blue dashed line indicates the most significant "bonding" noncovalent MBD mode, whereas the red dashed line indicates the corresponding most "anti-bonding" MBD mode. The insets represent the contributions of the most significant bonding and anti-bonding MBD modes to the interatomic MBD energy $V_{MBD}$. In the right panel **b**, the analogous analysis is reported for the "tweezer" complex with 1,4-dicyanobenzene, dominated by dispersion interactions. All the energies are expressed in eV. Source data are provided as a Source Data file.

clear advantage over its first-quantized counterpart in identifying noncovalent orbitals that significantly determine interactions in realistic chemical and biological systems.

## SQ-MBD analysis of collective plasmon-like modes
Finally, we demonstrate insights into electronic and quantum-information properties enabled by the SQ-MBD method. MBD transition dipoles are relevant quantities required to calculate the static and dynamic polarizabilities of the coupled QDOs.

Due to Fock-state selection rules, transition dipoles in a system of linearly coupled QDOs are allowed only between the MBD ground state $|\tilde{\mathbf{0}}\rangle$ and singly excited states, $\left|\tilde{\mathbf{1}}_k\right\rangle = \left|\tilde{0}_1,...,\tilde{1}_k,...\tilde{0}_{3N}\right\rangle$. The $x_i$th Cartesian component of the electric transition dipole between the ground state and the state with a single excitation in the $k$th MBD mode is given by $(\mu_{x_i})_{\tilde{\mathbf{0}}\tilde{\mathbf{1}}_k} = \sum_{A=1}^{N}\langle\tilde{\mathbf{1}}_k|\hat\mu_{A_{x_i}}|\tilde{\mathbf{0}}\rangle$. We introduce the scalar quantity

$$|\bar\mu_{\tilde{\mathbf{0}}\tilde{\mathbf{1}}_k}|^2 = \frac{1}{3}\sum_{A,B=1}^{N}\sum_{i=1}^{3}\mu_{\tilde{\mathbf{0}}\tilde{\mathbf{1}}_k;A_{x_i}}\mu_{\tilde{\mathbf{0}}\tilde{\mathbf{1}}_k;B_{x_i}}^{*} \qquad (7)$$

that we will refer to as the isotropic (orientationally averaged) square modulus of the transition dipole associated with the $k$th MBD normal mode and that can be interpreted as the contribution of a specific MBD normal mode to the isotropic polarizability of the system. In Fig. 4, the plot of $|\bar\mu_{\tilde{\mathbf{0}}\tilde{\mathbf{1}}}|$ as a function of the normalized eigenenergy $\hbar\omega_k/(\hbar\tilde\omega_1)$ (adimensionalized by the lowest MBD eigenenergy units) associated with the $k$th MBD normal mode for crambin is shown. The MBD normal modes in the range between $1-2.5\,\tilde\omega_1$ exhibit a rather high isotropic transition dipole moment ~6–8 D on average, of the same order of magnitude as the transition dipoles of organic fluorophores[47,49,71,72]. A single mode near $1.06\,\tilde\omega_1$ has a much larger transition dipole ~12 D, arising from a strong collective atomic response. In many practical applications, it is relevant to ascertain how strongly correlated are transition dipole elements between specific residues. With this aim in mind, we introduce the two-fragment contribution to the effective

square modulus of the isotropic transition dipole. In the insets of Fig. 4, plots of the matrix defined in Eq. (7) are reported for two MBD normal modes: the low-frequency mode with the highest transition dipole ($k = 13$) and the highest-frequency MBD mode ($k = 1926$) having a much smaller total transition dipole. The results show that the low-frequency mode strongly correlates dipole fluctuations over many residues of the whole molecule, while the high-frequency mode correlates fluctuations essentially of a single residue. The previous considerations and analysis can be extended to the calculation of higher-order transition multipoles.

## Quantum mutual information of atomic QDOs in the MBD ground state
In the previous sections, we have elucidated how the SQ-MBD framework facilitates the estimation of fragment contributions to MBD-related observables, such as MBD energies and transition dipoles within a given system partition. This suggests an MBD coarse-graining approach for large molecular systems by partitioning each system into fragments, including elements of a complex environment, subsequently employing a high-level methodology to compute the dynamical polarizability and MBD energy for each fragment and finally resolving fragment interactions through a lower-level method. This strategy draws a parallel with quantum-embedding techniques employed in electronic structure theory[69]. However, such a coarse-graining scheme raises the challenge of defining an optimal strategy for partitioning a quantum system composed of quantum degrees of freedom under the constraint of minimizing interfragment correlations. For this purpose, quantum information methods can be applied to study the correlations among QDO subsets in the MBD ground state.

Quantum information methods have recently found successful applications in studying electronic correlation properties, in particular for the construction of a distinctive class of correlation energy functionals in the reduced density matrix functional theory[73,74].

In the MBD framework, mutual information can be used to partition a given system into fragments that minimize inter-fragment MBD

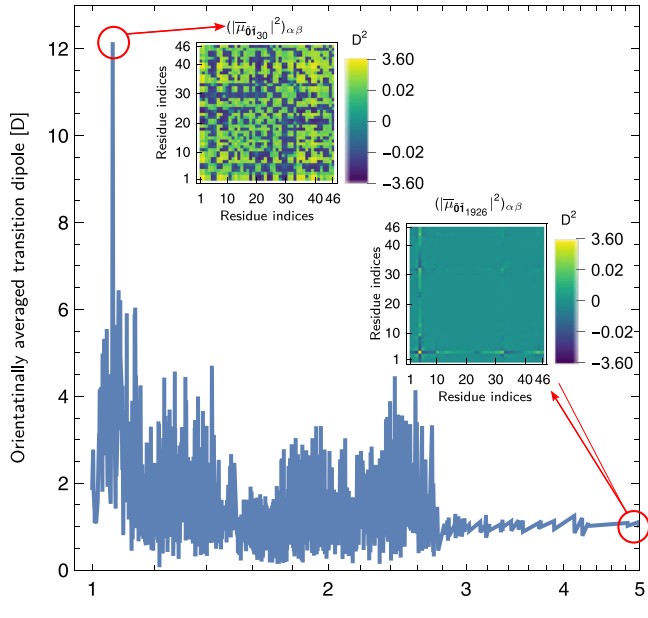

**Fig. 4 | Transition dipoles of the many-body dispersion (MBD) modes for crambin.** The figure illustrates the orientationally averaged transition dipole moment denoted as $|\bar{\mu}_{\tilde{0}\tilde{1}_k}|$ for the $k$th MBD eigenmode as a function of the corresponding MBD excitation energy $\hbar\tilde{\omega}_k$. The MBD energies on the $x$-axis are expressed in units relative to the lowest frequency eigenmode, i.e. $\hbar\tilde{\omega}_1$. In the insets, the matrix elements of the square of the orientationally averaged transition dipole for the interacting residues. In particular, for two fixed residues $\alpha$ and $\beta$, $(|\bar{\mu}_{\tilde{0}\tilde{1}_k}|^2)_{\alpha\beta} = 1/3 \sum_{A \in \mathcal{F}_\alpha, B \in \mathcal{F}_\beta} \sum_{i=1}^{3} \mu_{\tilde{0}\tilde{1}_k:A_{x_i}} \mu^*_{\tilde{0}\tilde{1}_k:B_{x_i}}$ are shown. Transition dipoles are expressed in Debyes. Source data are provided as a Source Data file.

interactions. In particular, the unitary representation of Bogoliubov transformations in Eq. (4) shows that the MBD ground state is a multimodal Gaussian state of the same type as the ones used in continuous-variable quantum information theory[75,76]. The quantum information-derived observable quantifying correlations among two parts of a given quantum system is the quantum mutual information, defined as

$$(\text{M.I.})_{\alpha\beta} = S[\hat{\rho}_\alpha] + S[\hat{\rho}_\beta] - S[\hat{\rho}_{\alpha\beta}], \qquad (8)$$

where $S[\rho_\alpha] = -\text{Tr}[\hat{\rho}_\alpha \log \hat{\rho}_\alpha]$ is the von Neumann entropy of the reduced density matrix $\hat{\rho}_\alpha = \text{Tr}_{\gamma \neq \alpha}(|\tilde{0}\rangle\langle\tilde{0}|)$, and $\hat{\rho}_{\alpha\beta} = \text{Tr}_{\gamma \neq \alpha, \beta}(|\tilde{0}\rangle\langle\tilde{0}|)$. The method used to evaluate the von Neumann entropy for a fragment $\mathcal{F}_\alpha$ relies on the symplectic spectrum of the correlation matrix $\Sigma^{(\alpha)}$

$$\Sigma^{(\alpha)} = \begin{pmatrix} \langle \hat{\boldsymbol{a}}_\alpha \otimes \hat{\boldsymbol{a}}_\alpha \rangle_{\tilde{0}\tilde{0}} & \langle \hat{\boldsymbol{a}}_\alpha \otimes \hat{\boldsymbol{a}}_\alpha^\dagger \rangle_{\tilde{0}\tilde{0}} \\ \langle \hat{\boldsymbol{a}}_\alpha^\dagger \otimes \hat{\boldsymbol{a}}_\alpha \rangle_{\tilde{0}\tilde{0}} & \langle \hat{\boldsymbol{a}}_\alpha^\dagger \otimes \hat{\boldsymbol{a}}_\alpha^\dagger \rangle_{\tilde{0}\tilde{0}} \end{pmatrix} \qquad (9)$$

(see refs. 75,76 and section VI of the Supplementary Discussion for further details). In the inset of the left panel of Fig. 5a, the results of the calculation of mutual information between pairs of residues in the MBD ground state are reported. Our analysis suggests that a strong connection exists between mutual information among QDOs in different residues and the interfragment contribution to the MBD potential energy: both quantities, in fact, are determined by the covariance matrix in Eq. (9). In particular, as expected, the mutual information between residues strongly correlates with the inter-residue distance $d$. Figure 5a reveals that MI scales as $\sim d^{-6.16 \pm 0.04}$, albeit with a substantial scatter. This scaling law follows the inter-residue vdW interaction energy; in fact, the mutual information between pairs of residues and the mutual interaction energies $V_{\text{MBD}}$ for the same pairs

are strongly correlated (see section IV of the Supplementary Discussion).

The inter-fragment mutual information matrix can also be used to construct graphs representing the correlations among quantum electronic density fluctuations represented by atomic QDOs. In fact, the mutual information matrix can be interpreted as the adjacency matrix of weighted graphs whose nodes are the fragments. Since the entries of the mutual information matrix are non-negative, it is possible to assign a centrality measure to each fragment using, for instance, the eigenvector centrality $s_\alpha$, defined as[77]

$$s_\alpha = \frac{1}{\lambda_{\max}^{(\text{MI})}} \sum_{\beta=1}^{N_{\text{frag}}} (\text{M.I})_{\alpha\beta} \, s_\beta, \qquad \sum_{\alpha=1}^{N_{\text{frag}}} \| s_\alpha \|^2 = 1, \qquad (10)$$

where $\lambda_{\max}^{(\text{MI})}$ is the largest eigenvalue of the mutual information matrix. In the context of our investigation, we present a case study involving the structural analysis of crambin partitioned into individual residues. The results of this analysis are depicted in Fig. 5b. Notably, our findings highlight phenylalanine (PHE13) as the most central residue, reaffirming its unique characteristics regarding its contribution to the MBD energy. Through quantum mutual information analysis, we explicitly unveil its significant interactions with the protein environment, as suggested by the analysis of the intra- and interfragment contributions to MBD energy previously discussed. Specifically, PHE13 exhibits coordination with residues situated on the same alpha-helix, such as the arginines (ARG10 and ARG17), as well as with residues on the second alpha-helix, notably threonine (THR30). This observation suggests a potentially central role played by PHE13 in shaping the protein's overall structure. Such an analysis will find applications in future studies on intra- and intermolecular allosteric pathways supported by quantum electronic density fluctuations, and could also be used to identify the criticality of single-point mutations for the structural integrity of de novo protein designs. Finally, a more advanced analysis can be developed in the same SQ-MBD framework, examining multifragment correlations among QDOs and generalizing mutual information concepts to multipartite-entangled systems[78,79].

## Discussion

In summary, we have presented a formulation of the MBD model in the second quantization picture (SQ-MBD), leading to computational and conceptual insights into coupled QDOs in intricate molecular systems. The presented method allowed us to investigate the ground state of the MBD Hamiltonian in terms of the superposition of QDO excited states. Owing to the Fock space representation in the SQ-MBD framework, it becomes possible to simplify the calculation of expectation values of observables in and between MBD ground and excited states. In fact, the SQ-MBD formalism makes clearer the connection not only between the non-interacting and the interacting ground state for the system of QDOs but also between the atomic QDO excitations and the MBD collective plasmon-like quasiparticle excitations.

The analysis based on the SQ-MBD approach combines well-known methods from different areas of physics spanning atomistic and field perspectives (including MBD, electronic structure theory, Bogoliubov transformations, field theory, and quantum information tools) into an original method of analysis for correlations among quantum electronic density fluctuations in large molecular complexes, a hitherto unsolved problem in achieving computational efficiency. In particular:

- SQ-MBD provides a suitable framework to compute and analyze the contribution of arbitrary sub-fragments to many important properties of the whole system, including the total MBD energy or the transition dipoles of MBD modes. This class of results opens an avenue for coarse-grained and field-excited MBD

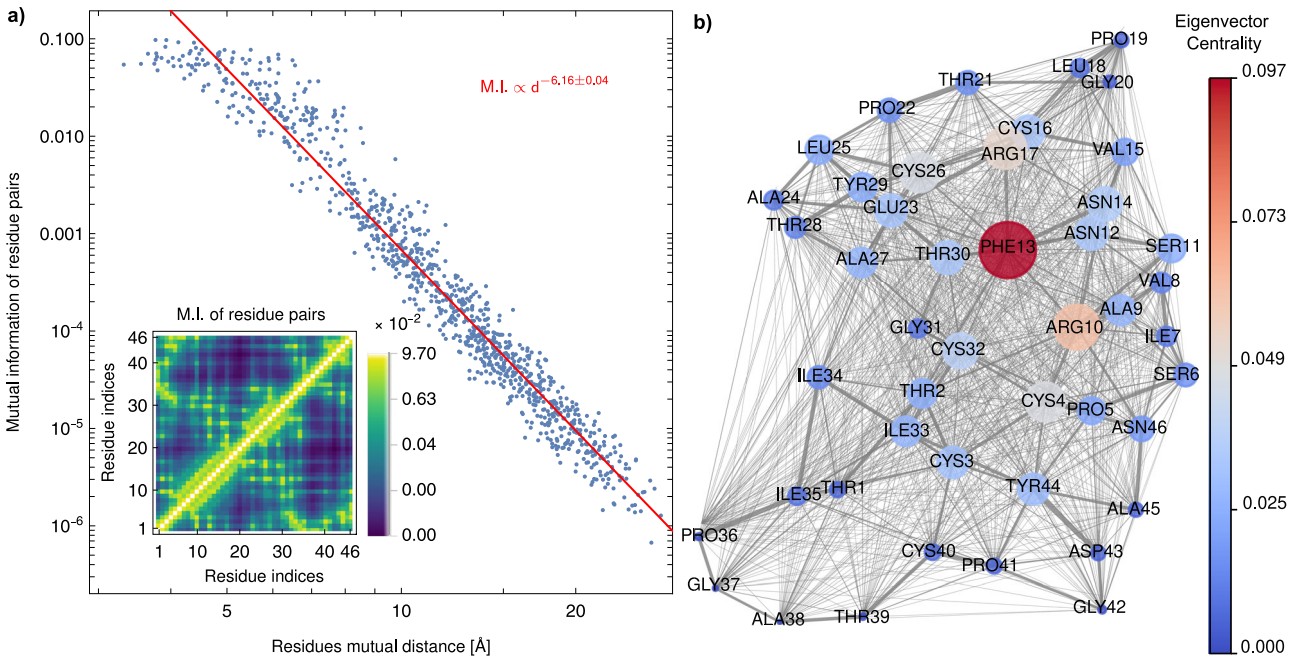

**Fig. 5 | Quantum information analysis of the MBD ground state in crambin.** Left panel **a** shows the scatter plot of the mutual information (MI) for pairs of residues vs. the distance $d$ between their centers of mass. Distances are expressed in angstrom. In the inset, the mutual information of each residue pair is shown. Right panel **b** shows the weighted graph whose adjacency matrix coincides with the mutual information matrix reported in the inset of panel a). The color and the size of each node are proportional to the eigenvector centrality of each residue. Source data are provided as a Source Data file.

models in the established area represented by research on many-body van der Waals interactions;

- SQ-MBD enhances our understanding of the properties of MBD modes that recent results suggest strongly correlate with exciton and optical modes in molecular systems[12], and it allows for equivalent treatments of the electromagnetic field and its interactions with intricate matter; and

- SQ-MBD provides a natural approach to connect the MBD model of coupled QDOs with quantum information theory, which can be used for the development of straightforward methods to analyze correlations and (non)separability among different fragments in complex molecular systems. These results thus represent the starting point for the development of computationally efficient strategies to enable the application of MBD interactions to molecular complexes with an even larger number of atoms.

Moreover, the possibility to easily compute matrix elements of higher-order terms in the multipole expansion of the Coulomb interaction potential among QDOs would be a first step for an extension of the method proposed in ref. 57 for the evaluation of MBD energy beyond the dipole–dipole approximation. The SQ-MBD approach could be extended to periodic systems akin to its parent MBD Hamiltonian[15], where cooperative effects among atomic QDOs (mediated by plasmon-like MBD modes) may provide enhanced insights owing to the high symmetry of Bloch states in such systems.

## Methods

The MBD method is currently built on top of a given DFT method and encodes properties of the ground state electronic density function in the parameters of QDOs, i.e. the frequencies $\omega_A$ and the polarizabilities $\mathcal{A}_A^{(0)}$. The parameterization of quantum Drude oscillators (QDOs) adopted in this paper is based on a two-step procedure. First, the polarizability $\mathcal{A}_A$ (expressed in volume units), the $C_{6,AA}$ dispersion coefficient, and the van der Waals radius $R_{\mathrm{vdW},A}$ of each atom $A$ in the

system were calibrated via the Tkatchenko–Scheffler method[80]

$$
\begin{cases}
C_{6,AA}^{(\mathrm{aim})} = \eta_A^2 C_{6,AA}^{(0)} \\
\mathcal{A}_A^{(\mathrm{aim})} = \eta_A \mathcal{A}_A^{(0)} \\
\mathcal{R}_{\mathrm{vdW},A}^{(\mathrm{aim})} = \eta_A^{1/3} R_{\mathrm{vdW},A}^{(0)}
\end{cases}
\tag{11}
$$

where $\eta_A = h_A/Z_A$ is the ratio between the on-site contribution $h_A$ to the Mulliken population (corresponding to the atom-projected trace of the density matrix) and the atomic charge $Z_A$ in the case of a free atom[81] applied to electronic structure calculations from density functional-based tight binding (DFTB) including electrostatics and Pauli repulsion. The estimation of the $\eta_A$ ratios is a key step in setting the parameters of the atomic QDOs. In order to refine the estimation of non-local effects for atom-in-molecule (AIM) response properties, the natural orbital functional theory framework could provide an alternative choice, taking into account the contribution to AIM electric response properties of the off-diagonal density matrix elements. The second step is the initialization of the MBD Hamiltonian and its diagonalization. The $C_{6,AA}^{(\mathrm{aim})}$ coefficient and the polarizabilities $\mathcal{A}_A^{(\mathrm{aim})}$ are rescaled according to the "range-separated self-consistent screenin" procedure (the 'rsscs' option in libMBD[82]) to account for electrodynamic screening using the short-range part of the range-separated dipole tensor for quantum harmonic oscillators (see ref. 83). In this way, the parameters $\mathcal{A}_A^{\mathrm{SCS}}, C_{6,AA}^{\mathrm{SCS}}, R_{\mathrm{vdW},A}^{\mathrm{SCS}}$ of each atom $A$ have been fixed, taking into account both the atom-in-molecule features and self-consistent screening. It has been shown that the MBD@rsSCS method could exhibit qualitative problems with ionic and hybrid metal-organic systems, as a poor prediction of ionic polarizabilities and the so-called phenomenon of polarization catastrophe in the transition-metal dichalcogenides. The fractional ionic approach for the polarizability of ions[84] and the MBD-NL method[85] have been proposed to solve these issues. We emphasize that the SQ-MBD formalism can be easily extended to the MBD-NL method without any particular modification. However, we do not expect qualitative differences for the specific

systems investigated in our manuscript between the more accurate MBD-NL method and the MBD@rsSCS method.

The dipole–dipole potential between a pair of three-dimensional QDOs is given by the rank-2 (3 × 3) tensor

$$\mathbb{T}_{AB}(\mathbf{R}_{AB}) = \nabla_{\mathbf{R}_A} \otimes \nabla_{\mathbf{R}_B} |\mathbf{R}_{AB}|^{-1} = \frac{(\mathbb{I} - 3\hat{R}_{AB} \otimes \hat{R}_{AB})}{|\mathbf{R}_{AB}|^3} \quad (12)$$

where $\mathbf{R}_{AB} = \mathbf{R}_B - \mathbf{R}_A$ is the distance vector between atoms $A$ and $B$. According to the MBD@rsSCS method, the dipole-dipole coupling tensor used in the MBD Hamiltonian is modified compared to the one in Eq. (12): for each couple of atomic QDOs, the tensor $\mathbb{T}_{AB}$ is multiplied with a function denoted as $g_{\mathrm{rs},AB}(\|\mathbf{R}_{AB}\|)$, dampening the potential at short distances. Such a modification of the dipole–dipole potential prevents the double-counting of dipole–dipole interactions at short distances, as these effects are already considered through the self-consistent rescaling of the QDOs parameters. Adopting the notation presented in ref. 83, the damping function for each atomic oscillator pair's range-separated potential was chosen to be

$$g_{\mathrm{rs},AB}(\| \mathbf{R}_{AB} \|) = 1 - \frac{1}{1 + e^{-a[\|\mathbf{R}_{AB}\|/(\beta R_{\mathrm{vdW},AB}^{(\mathrm{aim})})-1]}}, \quad (13)$$

where $R_{\mathrm{vdW},AB}^{(\mathrm{aim})} = R_{\mathrm{vdW},A}^{(\mathrm{aim})} + R_{\mathrm{vdW},B}^{(\mathrm{aim})}$ and the parameters $a = 6.0$ and $\beta = 0.83$ have been fixed. The characteristic angular frequency of the atom $A$ is given by

$$\omega_A = \frac{4}{3} \frac{C_{6,AA}}{\hbar \mathcal{A}_A^2} . \quad (14)$$

The diagonalization of the $3N \times 3N$ MBD Hamiltonian matrix was performed using libMBD software, thus obtaining the orthogonal transformation $\mathcal{O}$ between atomic QDOs and MBD normal modes, as well as the $3N$ eigenvalues $\tilde{\omega}_k$ corresponding to these MBD eigenmode frequencies.

The matrix $M(\boldsymbol{\omega})$ maps from the first- to second-quantized representation for the atomic QDOs and takes the form

$$M(\boldsymbol{\omega}) = \begin{pmatrix} M_{aq} & M_{ap} \\ M_{a^\dagger q} & M_{a^\dagger p} \end{pmatrix} = \frac{1}{\sqrt{2\hbar}} \begin{pmatrix} \mathcal{D}^{\frac{1}{2}}(\boldsymbol{\omega}) & i\mathcal{D}^{-\frac{1}{2}}(\boldsymbol{\omega}) \\ \mathcal{D}^{\frac{1}{2}}(\boldsymbol{\omega}) & -i\mathcal{D}^{-\frac{1}{2}}(\boldsymbol{\omega}) \end{pmatrix}, \quad (15)$$

where $\mathcal{D}(\boldsymbol{\omega}) = \mathrm{diag}(\omega_1, \omega_1, \omega_1, ..., \omega_N, \omega_N, \omega_N)$ is a $3N \times 3N$ diagonal matrix of QDO eigenfrequencies. $\tilde{M}(\tilde{\boldsymbol{\omega}})$ maps from first- to second-quantized representation for the MBD normal modes. The $X,Y$ matrices in Eq. (2) of the main text define the linear transformation between the creation/annihilation operator algebra for the atomic QDOs and the one for the MBD normal modes. These matrices can be expressed in terms of the orthogonal matrices $\mathcal{O}$ and the transformation matrices $M(\boldsymbol{\omega}), \tilde{M}(\tilde{\boldsymbol{\omega}})$:

$$X = \left[ \tilde{M}_{b\tilde{q}}(\tilde{\boldsymbol{\omega}})\mathcal{O}_{\tilde{q}q}M_{qa}^{-1}(\boldsymbol{\omega}) + \tilde{M}_{b\tilde{p}}(\tilde{\boldsymbol{\omega}})\mathcal{O}_{\tilde{p}p}M_{pa}^{-1}(\boldsymbol{\omega}) \right] = \frac{1}{2}\left[ \tilde{\mathcal{D}}^{1/2}\mathcal{O}\mathcal{D}^{-1/2} + \tilde{\mathcal{D}}^{-1/2}\mathcal{O}\mathcal{D}^{1/2} \right]$$

$$Y = \left[ \tilde{M}_{b\tilde{q}}(\tilde{\boldsymbol{\omega}})\mathcal{O}_{\tilde{q}q}M_{qa}^{-1}(\boldsymbol{\omega}) - \tilde{M}_{b\tilde{p}}(\tilde{\boldsymbol{\omega}})\mathcal{O}_{\tilde{p}p}M_{pa}^{-1}(\boldsymbol{\omega}) \right] = \frac{1}{2}\left[ \tilde{\mathcal{D}}^{1/2}\mathcal{O}\mathcal{D}^{-1/2} - \tilde{\mathcal{D}}^{-1/2}\mathcal{O}\mathcal{D}^{1/2} \right] . \quad (16)$$

### Reporting summary
Further information on research design is available in the Nature Portfolio Reporting Summary linked to this article.

### Data availability
The data generated in this study and reported in Figures and Supplementary Discussion are provided in the Source Data file. The molecular structures and the data generated in this study have been deposited in[86]. Source data are provided with this paper.

### Code availability
The software used to derive the CPA ratios required by MBD@rsSCS is an adapted version of the software package DFTB+[87] developed by M. Stöhr and available on Github repository[88]. The diagonalization of the MBD Hamiltonian was executed using the software library libMBD[82,89]. Python and Mathematica[90] scripts for data analysis and figure generation are available from the authors.

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

## Acknowledgments

M.G. and A.T. acknowledge support from the European Research Council (ERC Advanced Grant "FITMOL") and Fonds National de la Recherche Luxembourg (FNR CORE grant BroadApp C20/MS/14769845). M.G. would like to thank Mario Galante, Marco Pezzutto, and Martin Stöhr for their helpful suggestions. P.K. acknowledges support from the Alfred P. Sloan Foundation (Matter-to-Life program), the U.S. National Science Foundation (Quantum Leap Challenge Institutes and Research Coordination Network programs), the Whole Genome Science Foundation, and the U.S.–Italy Fulbright Commission. The authors would also like to acknowledge discussions at the Institute for Pure and Applied Mathematics, and valuable insights from Georgia Dunston, Marco Pettini, and Giuseppe Vitiello.

## Author contributions

M.G., A.T., and P.K. were responsible for the study's conception and design. M.G. conducted data collection, while M.G., A.T., and P.K. collectively performed the analysis and interpretation of results. The draft manuscript was prepared by M.G., A.T., and P.K. All authors participated in discussions and endorsed the final manuscript version.

## Competing interests

The authors declare no competing interests.
