## [Peer Review File · Nature Communications]

Second Quantization of Many-Body Dispersion Interactions for Chemical and Biological SystemsREVIEWER COMMENTS

Reviewer #1 (Remarks to the Author):

In this manuscript, the Authors develop a second-quantized many-body dispersion technique for calculating dispersion interactions for systems of many atoms and obtaining insights on the role of quantum fluctuations in molecular systems. This also yields a computational framework useful to treat such systems, and several examples are discussed in detail. The research subject is important in the current relative research framework.

The paper appears physically sound, original, and reasonably well written. It certainly deserves publication.

There are a few points that in my opinion should be addressed by the Authors before publication.

- In the model used, only dipole interactions are included (see Eq. (1) of the manuscript). The Authors should motivate in detail why higher multipolar interactions are not relevant in the specific physical systems they consider (and, possibly, state the conditions under which such approximation is reliable).
- It seems that only two-body dipolar interactions are included. van der Waals interactions, however, are not additive. Why, in the systems considered, many-body interactions can be neglected? Also this approximation should be properly motivated.
- At page 3 of the manuscript, a 3D isotropic QDO model has been chosen: does this mean that some rotational averaging is assumed? If so, this should be explicitly mentioned and physically motivated.
- At page 3 of the manuscript, at the end of the paragraph following Eq. (1), some considerations about the limitation of the MBD approach for excited collective states are mentioned; a few more words on this important point could improve the presentation of the paper and its readability, in particular for a nonspecialistic reader.
- The Authors state that their SQ-MBD technique overcomes the MBD approach in some relevant issues; I would suggest the Authors to explain and motivate this point in more details, possibly independently from specific examples/applications.
- The considerations about quantum-information properties, starting from the end of page 8 of the manuscript, are not very clear. Are they related to the presence of strongly correlated dipole fluctuations?
- The importance and the specific computational simplifications of the SQ-MBD approach presented in the manuscript should be introduced with a proper detail also in the paper, for example in the introduction (independently from the Supplementary Material).
- An optional point: in my opinion, in the main text of the paper, a sharper separation of the methods introduced (in their different aspects) and their application to the different systems considered could improve the readability of the paper, in particular for a nonspecialistic reader.

In my opinion, this paper can be accepted for publication in Nature Communications after that the Authors have addressed the points raised above.

Reviewer #2 (Remarks to the Author):

Gori, Kurian, & Tkatchenko ("Second Quantization Approach to Many-Body Dispersion...")

The authors present a second-quantized formulation of the MBD methodology that has been developed by Tkatchenko and coworkers over the past decade. The method is based on a form of coarse graining in which electrons on each atom are mapped onto a 3D (isotropic) harmonic oscillator, with dipole coupling between these Drude-type oscillators, diagonalizing the coupled-oscillator Hamiltonian to obtain the dispersion energy. This technique, in its "MBD@rsSCS" version (used here) has been extremely successful in the context of density functional theory (DFT), although MBD is distinct from DFT.

The main theoretical development in the present work is a mapping from the atomic oscillator

representation (which the authors call "first quantization") to the normal modes of the dipole coupling matrix that defines the potential energy term in the Hamiltonian. This is a straightforward exercise and in fact the eigenmodes of that matrix have been examined elsewhere by Tkatchenko and coworkers in order to gain physical insight from MBD calculations [Nat. Commun. 8, 14052 (2017), oddly not cited here]. I see the present work as largely duplicative without much that's new; it is largely just visualization of the eigenmodes of the MBD Hamiltonian. Although the authors have much to say about how the eigenmode representation ("second quantization") will be useful for obtaining physical insight in nano- or mesoscale systems, or for providing inputs for machine learning force fields, all of that discussion is rather speculative and none of it is demonstrated in the present work. Therefore, it remains unclear to this reviewer how the second-quantized formulation goes much beyond the kind of analysis that has already been possible with MBD. It was always possible to examine atomic contributions to the eigenmodes of the harmonic oscillator coupling matrix, and partitioning those contributions is the type of additional coarse-graining that is suggested here. The work does provide new examples of how this partitioning might look for various molecular systems including a small protein but I don't see that a lot of physical insight is gained.

A couple of other comments:

(*) One thing that's missing is any discussion of the interplay of intermolecular forces besides dispersion. For example, there is a discussion of how the asymmetric geometry of the C70-CPPA complex in Fig. 1b manifests as asymmetry in the oscillator excitations. However, that asymmetry is probably not predictable from dispersion alone, but results from competition with Pauli repulsion and electrostatics in whatever (probably DFT) calculation was used to obtain the geometry. Such a detailed analysis of dispersion, without considering how other forces contribute to the behavior dispersion, seems itself to be somehow asymmetric.

(*) The authors comment on the sizable magnitude of the total dispersion energy, larger than a covalent bond energy in some cases, which must contribute to "driving the dynamics of biomolecular systems". Dispersion is extensive with system size so it will always be larger than a covalent bond strength if the molecule is large enough but dispersion is also non-specific and not directional unlike some other intermolecular forces such as H-bonding or dipolar electrostatics. Therefore, I don't think that simply noting the magnitude of dispersion suffices to demonstrate its importance as a driving force for any particular biomolecular process.

Overall, I feel that this material is quite technical and does not significantly advance the MBD formalism beyond what has already been done by Tkatchenko and coworkers. For those reasons, I feel that this manuscript would be more suitable for a more specialized journal.

Reviewer #3 (Remarks to the Author):

see the attached pdf file report.pdf

Manuscript ID: NCOMMS-23-15403

TITLE: Second Quantization Approach to Many-Body Dispersion Interactions: Implications for Chemical and Biological Systems

AUTHORS: Matteo Gori, Philip Kurian, Alexandre Tkatchenko

In the submitted manuscript, the authors propose a novel methodological framework, SQ-MBD, for the analysis of many-body dispersion interactions, as emerging from the widely used state-of-the-art approach MBD@rsSCS. The proposed framework is based on sound physical theory and provides access to several interesting pieces of information involving contributions of individual atoms to the dispersion energy and importance of collective effects. The SQ-MBD method also enables one to partition the dispersion energy into components from arbitrarily defined structural fragments and to make an evaluation of their mutual pairwise interactions. Several types of SQ-MBD-based analyses are exemplified in real-world cases, including C_{70} fullerene surrounded by a large molecular ring, crambin protein, and selected systems from the S12L benchmark set. The manuscript is well written, the scientific presentation is sound, and the practical examples presented provide a clear demonstration of the usefulness of the approach. Overall, I am convinced that the submitted work represents a substantial contribution to the field of computer simulations of molecular and extended systems and deserves to be published in Nature Communications, provided the comments listed below are carefully addressed.

- 1.) The original formulation of the MBD method, as presented in J. Chem. Phys. 138, 074106 (2013) allows a rather elegant and computationally inexpensive way the partitioning of MBD correlation energy into individual atomic contributions. As described in Sec. II of the above-mentioned work, the MBD energy can be expressed as

$$E_{MBD} = -\frac{1}{2\pi} \int_0^\infty d\omega \sum_{n=2}^{\infty} \frac{1}{n} \text{Tr}\{(AT)^n\}$$

from which the following expression for the contribution of an atom A to the energy can be deduced:

$$E_{MBD,A} = -\frac{1}{2\pi} \int_0^\infty d\omega \sum_{n=2}^{\infty} \frac{1}{n} \text{Tr}\{(AT)^n\}_{A,A}$$

where $\{(AT)^n\}_{A,A}$ denotes a 3×3 block of the AT matrix (as defined in J. Chem. Phys. 138, 074106 (2013)). If I understand correctly, this piece of information should be identical to $(U_{MBD})_A$ (eq. 11 in SM). This alternative route to this quantity should be mentioned in the paper and, ideally, consistency of both approaches should be demonstrated.

- 2.) The physical model behind MBD@rsSCS suffers from some serious issues not mentioned in the manuscript, which preclude a direct application of the method to some of the emblematic vdW systems, such as the transition metal dichalcogenides (see, e.g., Sec. 2.3 of JCTC 12, 5920 (2016)). In the meantime, some of the authors of the submitted work introduced an improved version of MBD, called MBD-NL (PRL 124, 146401 (2020)), which is presumably free of such problems. Why did the authors choose to develop the SQ-MBD approach based on the MBD@rsSCS framework, if MBD-NL “increases the accuracy and efficiency of existing vdW functionals and is shown to be broadly applicable to molecules, soft and hard materials including ionic and metallic compounds, as well as interfaces between organic molecules and inorganic materials.”? This is very confusing because, to my understanding, the MBD-NL approach is meant to be the most accurate MBD-based approach available to date and as such it should eventually replace all the other MBD flavours. A comment on that would be useful. In any case, it would be fair if the problems of the MBD@rsSCS are mentioned in the paper.
- 3.) A minor point – I guess the last sign in the first line of the eq. 8 in SM should be “-” instead of “+”

**Answer to Referees of NCOMMS-23-15403 *Second Quantization
Approach to Many-Body Dispersion Interactions:
Implications for Chemical and Biological Systems***

Matteo Gori, Philip Kurian, and Alexandre Tkatchenko

(Dated: July 2023)

REVIEWER #1

General consideration of the referee *In this manuscript, the Authors develop a second-quantized many-body dispersion technique for calculating dispersion interactions for systems of many atoms and obtaining insights on the role of quantum fluctuations in molecular systems. This also yields a computational framework useful to treat such systems, and several examples are discussed in detail. The research subject is important in the current relative research framework. The paper appears physically sound, original, and reasonably well written. It certainly deserves publication. There are a few points that in my opinion should be addressed by the Authors before publication.*

Answer We thank the referee for appreciating the manuscript.

Point 1) *In the model used, only dipole interactions are included (see Eq. (1) of the manuscript). The Authors should motivate in detail why higher multipolar interactions are not relevant in the specific physical systems they consider (and, possibly, state the conditions under which such approximation is reliable).*

Answer We agree with the referee on the necessity of providing further details on the choice of the dipole-dipole coupling approximation in the MBD Hamiltonian. The mathematical correspondence between the many-body dispersion (MBD) energy obtained from the Hamiltonian model of mutually interacting quantum dipoles (QDOs) and the correlation energy derived within the framework of the Adiabatic Connection Fluctuation Dissipation in Random Phase Approximation (ACFD-RPA) has been rigorously and explicitly proven for dipole-dipole interactions among QDOs (see Tkatchenko et al., *J. Chem. Phys.*, **138** (7), 2013). Our choice of coupling is mainly motivated by making consistent these two well-established methods for the calculation of many-body dispersion interactions.

Studying the effects of higher-order interactions among QDOs in multipolar expansion in the ACFD-RPA framework, and the self-consistent inclusion of such contributions within the MBD (Hamiltonian) model, is the object of ongoing research (see Massa, D. et al., *J. Chem. Phys.*, **154** (22), 2021, and Poier, P. P. et al., *J. Phys. Chem. Lett.*, **14** (6), 2023). Stöhr et al. (*Nat. Comm.* **12** (137), 2021) showed that the contributions of higher-order multipolar terms have been evaluated in the MBD ground state of QDOs mutually interacting through the dipole-dipole potential. The MBD@rsSCS (range-separated self-consistent screening)

framework used in that paper shares similarities with the one adopted in our manuscript: both utilize an MBD Hamiltonian model with a range-separated interaction potential among the QDOs, realized through the incorporation of a smoothed step function taking values near zero for interatomic distances R_{AB} smaller than a length scale λ_{AB} characteristic of each pair of atoms, and near one for $R_{AB} \gtrsim 1$. In the paper by Stöhr et al., it has been stated that a “rule-of-thumb” to estimate the strength of multipolar effects is the level of distortion of the QDOs charge distribution, i.e., when the 3D isotropy of the vacuum is substantially perturbed. In that work, it has been shown that multipolar effects are negligible for small biological dimers; as multipolar effects are expected to affect the interaction only on short distances, it can thus be safely assumed that they can be neglected when applied to larger biomolecular complexes.

In highly polarizable supramolecular systems, such as the one considered in Fig.1 of our work, the QDOs charge displacement in the MBD ground state is stronger, and the multipolar corrections have been calculated to account for 4 – 6% of the binding energy, while the dipole-dipole corrections to MBD account for 30 – 40%. Thus, multipolar terms are estimated to give a contribution to the total many-body dispersion energy of the order, at most, of 15 – 20% of the MBD dipole-dipole energy.

However, it should be stressed that our manuscript focuses on introducing the second-quantization formulation of MBD interactions and the new conceptual insights that can be obtained by adopting such a perspective, rather than improving the accuracy of the existing MBD@rsSCS method. In this respect, the SQ-MBD method could provide an optimal framework for evaluating more accurately multipolar effects using Rayleigh-Schrödinger perturbation theory. A detailed evaluation of higher-order multipolar contributions to the MBD energy remains beyond the scope of the present work.

Actions In the subsection “Results-The MBD Model”, we have specified the limitations of including only the dipole-dipole interaction energy in the MBD Hamiltonian for the supramolecular systems presented in our work. Moreover, we have briefly discussed how the SQ-MBD framework can be optimal for developing new strategies for the evaluation of higher-order multipolar effects based on the SQ-MBD formalism.

Point 2) *It seems that only two-body dipolar interactions are included. Van der Waals interactions, however, are not additive. Why, in the systems considered, many-body interac-*

tions can be neglected? Also this approximation should be properly motivated.

Answer We thank the referee for allowing us to clarify this crucial point. We want to emphasize that *many-body dispersion interactions have not been neglected in our work*. The MBD model describes how many-body, non-additive dispersion interactions among atoms arise as an effective potential in the nuclear coordinates, once the “faster” quantum Drude oscillator (QDO) degrees of freedom are traced out in the MBD Hamiltonian, a consequence of the Born-Oppenheimer approach inherent in the current formulation of the MBD model.

Although the MBD Hamiltonian operator \hat{H}_{MBD} contains just pairwise dipole-dipole coupling among QDOs, the expectation value in its ground state, obtained by tracing out the QDO degrees of freedom in the MBD interacting ground state, results in a highly nontrivial function of the atomic positions that cannot be written in general as the sum of pairwise contributions, i.e.,

$$E_{\text{MBD}}(\mathbf{R}) = \langle \tilde{\mathbf{0}}(\mathbf{R}) | \hat{H}_{\text{MBD}}(\hat{\mathbf{q}}, \hat{\mathbf{p}}, \mathbf{R}) | \tilde{\mathbf{0}}(\mathbf{R}) \rangle \neq \sum_{A \neq B} f_2(\mathbf{R}_A, \mathbf{R}_B). \quad (1)$$

In this sense, the van der Waals dispersion energy among atoms derived from the MBD framework is inherently many-body and not additive, as correctly pointed out by the reviewer.

Actions A paragraph has been added to the manuscript at the end of the subsection “Results-The MBD model” clarifying how many-body, non-additive interactions arise from the MBD model, especially for non-specialist readers.

Point 3) *At page 3 of the manuscript, a 3D isotropic QDO model has been chosen: does this mean that some rotational averaging is assumed? If so, this should be explicitly mentioned and physically motivated.*

Answer An isotropic parametrization for the 3D atomic QDOs in the MBD Hamiltonian is equivalent to assuming the isotropy of the atom-in-molecule polarizability tensor after the self-consistent screening procedure has been applied. This approximation is a standard in the different flavors of the MBD method usually adopted, and only reflects isotropy in each non-interacting atomic QDO. Anisotropy in the MBD eigenmodes is a given and arises naturally from interacting atomic QDO aggregates in (supra)molecular complexes, which of course exhibit rotationally asymmetric structures.

Nothing would prevent, in principle, the consideration of anisotropic QDOs obtained by

the self-consistent screening procedure. However, as discussed by Di Stasio et al., *J. Phys.: Condens. Matter* **26** (2014), this would require a matching anisotropic definition for the range-separated Coulomb potential among QDOs in the long-range regime. In order to obtain an accurate estimation of the MBD energy from a given DFT ground state, a range-separated potential is required to avoid double-counting the dispersion energy over short interatomic distances. The hypothesis of an isotropic atomic response yields the assumption that the step function included in the potential for the range-separation depends only on the interatomic distance. Including anisotropic QDO polarizability would yield an anisotropic step function for the range separation, i.e., depending on both the direction and modulus of the interatomic distance vector. Although a study of the effects of considering anisotropic QDOs in the non-interacting state would be of certain interest, this is beyond the scope of the present work.

Actions A footnote has been added in the subsection “Results-The MBD model”, in order to clarify the reasons for restricting our consideration to isotropic QDOs in the non-interacting Hamiltonian.

Point 4) *At page 3 of the manuscript, at the end of the paragraph following Eq. (1), some considerations about the limitation of the MBD approach for excited collective states are mentioned; a few more words on this important point could improve the presentation of the paper and its readability, in particular for a nonspecialistic reader.*

Answer We agree with the referee on the importance of clarifying this point.

Actions The discussion in the subsection “Results-Second Quantization formulation of the MBD method (SQ-MBD)” about the limitations of the first quantization perspective in describing the connection between atomic and collective degrees of freedom has been enlarged to stress the importance of a quasiparticle description of MBD modes for nonspecialist readers.

Point 5) *The Authors state that their SQ-MBD technique overcomes the MBD approach in some relevant issues; I would suggest the Authors explain and motivate this point in more details, possibly independently from specific examples/applications.*

Answer We welcome and take up the recommendation of the referee. It should be noticed that the central target of the MBD method (i.e., the non-explicit connection between quanta

of localized and collective fluctuations of the QDOs charge density) is strictly connected with the previous point raised by the referee, on the limitations of the first-quantized MBD approach for excited collective states.

In general, the second quantization formalism offers a Fock-space representation of the interacting MBD ground state by utilizing the unitary representation of the Bogoliubov transformation. Such a representation greatly simplifies the evaluation of matrix elements of observables that are polynomial in displacements and/or momenta of atomic QDOs, calculated in arbitrary eigenstates of the MBD Hamiltonian. Such calculations were impossible in the first-quantized formulation of MBD, and thus restricted the theoretical utility of the approach for comparison with experimental observations (e.g., excited-state transitions). By reducing the evaluation of Gaussian integrals over $3N$ real dimensions to matrix products involving Bogoliubov matrices and sets of selection rules, the connection between the degrees of freedom of atomic QDOs and the collective quasiparticle excitations of MBD becomes apparent, and makes feasible the treatment of electromagnetic field interactions with intricate matter, as described by MBD.

Furthermore, this approach facilitates the application of techniques from other research fields, such as quantum information, where the Bogoliubov representation can be used to identify strongly correlated subsystems in a multipartite-entangled network. The outcomes of these applications are particularly significant, as they provide valuable information for guiding coarse-graining schemes in MBD interactions. For instance, new physical insights can be extracted analyzing the contributions of various sub-fragments to essential properties of the entire system (e.g., the total MBD energy or the transition dipoles of MBD modes), or the contributions of specific sub-fragments to multipartite entanglement in complex and intricate systems. Indeed, ongoing and future work—beyond the scope of the present manuscript—will entail novel prediction of correlated subsystems (residues) as targets for ultrafast spectroscopy experiments in (bio)chemical systems. Moreover, we stress that the advantage of SQ-MBD over MBD clearly emerges in specific applications that require decoding information contained in the correlations among QDO fluctuations in the MBD ground and excited states.

Actions We have reported a summary of our answer to the referee in the subsection “Results-Second Quantization formulation of the MBD method (SQ-MBD)”. Additionally, a bullet list has been introduced in the conclusions to make clearer the applications where

the proposed SQ-MBD framework is more well-suited than the first-quantized method in providing physical, chemical, and biological insight on the role played by quantum electronic fluctuations, represented by the system of coupled QDOs, in (bio-, supra-)molecular complexes.

Point 6) *The considerations about quantum-information properties, starting from the end of page 8 of the manuscript, are not very clear. Are they related to the presence of strongly correlated dipole fluctuations?*

Answer We welcome the referee’s suggestion regarding the need for further clarification on our quantum information-based analysis of the MBD ground state. Our study focuses on quantifying the separability between arbitrary subsets (i.e., molecular “fragments”) of atomic QDOs in the MBD ground state. Our primary objective was to develop a methodology to assess the effectiveness of a given system partition (e.g., in amino acid residues or α -helices for proteins, or supramolecular units like the C70 ball) in achieving some weakly correlated fragments, thereby validating the reliability of treating those fragments in the given partition as isolated subsystems. A similar approach is taken in many quantum embedding approaches that are widely used in quantum information science. Indeed, weakly correlated fragments are distinguished by the presence of much stronger intra-fragment QDO correlations, even though all atomic QDOs are coupled pairwise through the dipole-dipole potential. We analyzed these correlations among QDOs located in different residues within the protein (crabbin) configuration at equilibrium. We found that the inter-fragment MBD potential energy serves as the primary information carrier for the separability of two residues in the MBD ground state. This key finding emphasizes that the strength of MBD inter-fragment interactions mainly determines the degree of separability of fragment pairs within the MBD ground state.

Actions The quantum information analysis in the subsection “Results-Mutual information of atomic QDOs in the MBD ground state” has been partially rewritten to improve its clarity.

Point 7) *The importance and the specific computational simplifications of the SQ-MBD approach presented in the manuscript should be introduced with a proper detail also in the paper, for example in the introduction (independently from the Supplementary Material).*

Answer We thank the referee for this suggestion.

Actions A Methods section has been added to the main text to include some parts of the Supplementary Material, specifically concerning the significance of and simplifications associated with the SQ-MBD approach.

Point 8) *An optional point: in my opinion, in the main text of the paper, a sharper separation of the methods introduced (in their different aspects) and their application to the different systems considered could improve the readability of the paper, in particular for a nonspecialistic reader.*

Answer We agree with the referee on this point.

Actions The structure of the paper has been completely reshaped, with a sharper separation in the sections (Introduction, Results, Conclusions, Methods) and subsections in the Results, to highlight the Methods introduced and their application to the different systems considered.

REVIEWER #2

Point 1a) *The main theoretical development in the present work is a mapping from the atomic oscillator representation (which the authors call "first quantization") to the normal modes of the dipole coupling matrix that defines the potential energy term in the Hamiltonian. This is a straightforward exercise and in fact the eigenmodes of that matrix have been examined elsewhere by Tkatchenko and coworkers in order to gain physical insight from MBD calculations [Nat. Commun. 8, 14052 (2017), oddly not cited here]. I see the present work as largely duplicative without much that's new; it is largely just a visualization of the eigenmodes of the MBD Hamiltonian.*

Answer Shifting from the first quantization to the second quantization formalism does not entail a simple mapping from the atomic oscillator representation to the collective normal MBD modes. Instead, it corresponds more precisely to a mapping from the displacements/momenta representation to a Fock space representation of the quantum system, as depicted in Figure 1a) of our manuscript. Although there is a correspondence between the first- and second-quantization descriptions of many-particle quantum systems, it is important to note that the information that can be readily accessed through these two approaches is not equivalent. This disparity has been extensively exploited in the realm of many-body quantum physics for characterizing collective phenomena, i.e., providing the ideal framework to construct quasiparticle representations.

Specifically, the original formulation of MBD is more suited for describing the charge distribution of atomic QDOs in real space, while the SQ-MBD framework allows representation of the MBD ground (and excited) states in terms of a superposition of atomic QDO excited states, i.e., the plasmon-like excitations of MBD collective modes are described in terms of bosonic QDO excitations. As mentioned in our answer to Point 5 of Reviewer #1, this SQ-MBD representation greatly simplifies the evaluation of matrix elements of observables that are polynomial in displacements/momenta (or, equivalently, creation/annihilation operators) of the atomic QDOs, calculated in arbitrary eigenstates of the MBD Hamiltonian.

In the paper mentioned by the referee [Nat. Comm. 8, 14052 (2017)], the MBD eigenmodes are represented as a charge density over QDOs in real space; i.e., the analysis relies on the first quantization formalism. Conversely, in our paper (see Figures 1 and 3), the central aspect remains the analysis of correlations among atomic QDO excitations in the

MBD ground and excited state using a *Fock space (second quantization) representation*. Furthermore, the referenced paper primarily focuses on investigating the contributions of MBD eigenstates on the binding energies of biomolecular complexes, defined as the difference of the interaction energy between two system configurations: one with the guest-host complex in the bound steady state (interacting), and the other with the two structures positioned at infinite separation (non-interacting). In contrast, our study concentrates on the comprehensive analysis of the distribution of MBD interaction energies between intra- and interfragment components within a specific partitioning scheme of a given system configuration of QDOs (see Figure 2). The binding energies (separated molecules vs. fully coupled molecules) that one obtains in these two ways are inequivalent as we discuss in our paper. Additionally, we examine the individual contributions of each MBD mode to this repartitioning of energy. By providing this comparison of our analysis with the investigation highlighted by the referee, we assert that the present analysis in our work is *original and non-duplicative* with respect to the existing literature on the MBD model.

Actions To avoid any confusion about this important point, we have added some additional explanation from our response above to the Introduction section.

Point 1b) *Although the authors have much to say about how the eigenmode representation ("second quantization") will be useful for obtaining physical insight in nano- or mesoscale systems, or for providing inputs for machine learning force fields, all of that discussion is rather speculative and none of it is demonstrated in the present work. Therefore, it remains unclear to this reviewer how the second-quantized formulation goes much beyond the kind of analysis that has already been possible with MBD. It was always possible to examine atomic contributions to the eigenmodes of the harmonic oscillator coupling matrix, and partitioning those contributions is the type of additional coarse-graining that is suggested here. The work does provide new examples of how this partitioning might look for various molecular systems including a small protein, but I don't see that a lot of physical insight is gained.*

Answer We agree with the referee that first- and second- quantization descriptions of quantum systems are related. However, the adoption of the second quantization formalism reveals striking advantages, both from a conceptual and computational point of view. The SQ-MBD approach employs a language well-suited for advancing, extending, and generalizing the treatment of complex quantum systems. For instance, in SQ-MBD, the electric

response of atom-in-molecule (AIM) properties can be modeled by M -level quantum systems instead of quantum Drude oscillators, thereby reducing the Hilbert space dimensionality of the problem. Moreover, the SQ-MBD model provides a quasiparticle description of MBD plasmon-like modes, thus allowing expression of the MBD ground (and excited) state(s) in terms of occupation numbers of atomic QDO non-covalent “orbitals”. Such a description enormously enhances the possibility of reading out the information about correlations among quantum electronic fluctuations in the MBD ground state; for instance, the Bogoliubov transformation specified in Eqs. (3) and (4) in the main text can be used to evaluate the contributions given by each process (creation/annihilation of excitons) of individual atomic QDOs to global observables in the MBD ground state, at any order in the atomic QDO creation/annihilation operators. We remark that this step of connecting the atomic level of description with a collective (global) one represented by the MBD modes is crucial in developing a coarse-grained model of MBD interactions suitable for the accurate and efficient description of million-atom systems, which are currently beyond existing MBD implementations. In fact, the main implications of the SQ-MBD projection of global observables onto intra- and inter-fragment contributions are extremely relevant for chemistry- and biology-oriented applications (as presented in Figure 2), and where “quantum embedding” approaches may be required for analyzing highly multipartite-entangled networks.

In addition, from a practical point of view, the SQ-MBD formulation allows one to efficiently compute relevant observables (not considered previously to this present work) and express any MBD excited state in terms of the excited states of atomic QDOs. An example of this is provided by the single- and pair-fragment contribution to the square modulus of the transition dipole associated with a given MBD mode (see Figure 3 in the main text). Such a quantity is particularly relevant in order to visualize and investigate the emergence of a network of correlations among local electronic fluctuations mediated by a given MBD mode excitation. In general, we think that a novel and original approach to the study of intermolecular interactions could emerge from the possibilities offered by SQ-MBD, to associate a network of correlations among fragments in large molecular systems to the MBD ground state (as in the case of MBD energy contributions or the mutual information between fragments) or to a given MBD single-excitation (as in the case of electric transition dipoles). The SQ-MBD framework offers a more inherent structure for the application of techniques developed in diverse fields of quantum physics, such as many-body quantum physics and

quantum information, to investigate quantum electronic fluctuations. The present work is intended to be the first step toward a reformulation of the theory of MBD interactions based on geometric information methods, which would constitute a potential breakthrough both from a fundamental point of view in establishing an entirely new perspective on MBD interactions and from the computational point of view in terms of efficiency in computing observables from MBD ground and excited states for large (up to million-atom) systems. Moreover, the insights that can be obtained from SQ-MBD analysis have relevance in other domains beyond the physical one, as stressed by the title.

Actions The new insights that are more easily accessible within the SQ-MBD framework have been recapitulated and emphasized in the conclusions.

Point 2) *One thing that’s missing is any discussion of the interplay of intermolecular forces besides dispersion. For example, there is a discussion of how the asymmetric geometry of the C70-CPPA complex in Fig. 1b manifests as asymmetry in the oscillator excitations. However, that asymmetry is probably not predictable from dispersion alone but results from the competition with Pauli repulsion and electrostatics in whatever (probably DFT) calculation was used to obtain the geometry. Such a detailed analysis of dispersion, without considering how other forces contribute to the behavior dispersion, seems to be somehow asymmetric.*

Answer We agree with the referee that the interplay of MBD interactions with other interactions, such as electrostatic interactions and Pauli repulsion, is a fundamental aspect for understanding both the steady-state configuration and non-steady-state configuration of molecular and supramolecular systems. We also affirm, as detailed in the Methods and Supplementary Material, that the atom-in-molecule polarizabilities upon which MBD is based are indeed derived from DFT calculations that include electrostatics and Pauli repulsion. However, MBD provides a quantification of the contributions due to long-range electrodynamic response that are not included from DFT calculations, i.e., the “ripple” on top of the primary “waves” of the molecular electronic density.

As specified in the Supplementary Material, the parametrization of the atomic QDOs, at the very core of the MBD method, derives from DFT calculations for predicting atom-in-molecule polarizability and C_6 coefficients. This implies that the MBD method is currently built *on top* of a given DFT method and encodes properties of the ground state electronic

density function. On the other hand, the MBD method provides a quantification of the contributions due to *long-range correlation* of electronic density fluctuations that are not included from semilocal DFT calculations.

However, in our manuscript, we have intentionally focused our analysis solely on the long-range MBD interactions and the projection of MBD observables onto molecular fragments, specifically considering *a fixed geometry of the nuclei in the structure*. Our objective was to investigate how the given nuclear configuration within a molecule influences the properties of MBD modes and the corresponding (projected) observables. While we acknowledge the relevance of exploring the interplay between MBD interactions and other intra- and intermolecular interactions, we must clarify that such an investigation goes beyond the scope of our present work. Nonetheless, we appreciate the referee’s suggestion, as it represents an intriguing application of the SQ-MBD method, which could be explored in future research.

Actions We have emphasized and recapitulated points from our response above in the Methods section.

Point 3) *The authors comment on the sizable magnitude of the total dispersion energy, larger than covalent bond energy in some cases, which must contribute to "driving the dynamics of biomolecular systems." Dispersion is extensive with system size so it will always be larger than a covalent bond strength if the molecule is large enough. Still, dispersion is also non-specific and not directional, unlike some other intermolecular forces such as H-bonding or dipolar electrostatics. Therefore, I don't think that simply noting the magnitude of dispersion suffices to demonstrate its importance as a driving force for any particular biomolecular process.*

Answer The computational significance of dispersion interactions in driving the dynamics of biomolecular systems has been recently investigated in recent studies comparing the outcomes of molecular dynamics simulations using both pairwise and full many-body van der Waals interactions. These investigations encompassed a toy model of biopolymers as well as small proteins (see Galante M. et al. *Phys. Rev. Res.*, **5**(1), L012028 and Unke, O. T. et al. *arXiv:2205.08306*). The findings from these works demonstrated that the MBD potential induces substantial qualitative alterations in the process of protein folding, when compared to molecular dynamics predicted using a pairwise description.

The referee has identified hydrogen bonding and dipolar interactions as instances of spe-

cific and directional interactions, in contrast to dispersion interactions. Regarding the concept of “directionality” in interactions, we assume that the referee refers to a potential that depends solely on the magnitude of the distance between atoms. In this context, the conventional pairwise van der Waals interaction potential lacks directionality, while the MBD potential exhibits directionality due to its general and highly nonlinear dependence on atomic positions. This characteristic is particularly evident in the Axilrod-Teller-Muto potential, an approximation of the MBD potential for three bodies. The Axilrod-Teller-Muto potential, derived from the third-order perturbation correction to the London dispersion energy, relies not only on the relative distances among three atoms but also on the angles formed by the mutual distance vectors, thereby demonstrating directionality as specified earlier.

In terms of interaction specificity, it is important to emphasize that the resonance or near-resonance condition in a system of quantum harmonic oscillators (QHOs) enhances correlations among these oscillators. The resonance condition among QHOs can be interpreted as a partial selectivity condition among them, and are manifested in the QDO networks described by MBD.

Finally, an examination of the coarse-grained MBD interactions in crambin reveals that certain residues exhibit significantly stronger inter-fragment interactions compared to others, thereby exhibiting a selectivity condition. Furthermore, the interaction energy among residues is not isotropic, further indicating the presence of directionality.

Actions We have added some explanation about the directionality of MBD interactions, recapitulated from our response above, at the end of Results-Local contribution to MBD interaction energy in biomolecular complexes.

REVIEWER #3

General consideration of the referee *In the submitted manuscript, the authors propose a novel methodological framework, SQ-MBD, for the analysis of many-body dispersion interactions, as emerging from the widely used state-of-the-art approach MBD@rsSCS. The proposed framework is based on sound physical theory and provides access to several interesting pieces of information involving contributions of individual atoms to the dispersion energy and importance of collective effects. The SQ-MBD method also enables one to partition the dispersion energy into components from arbitrarily defined structural fragments and to make an evaluation of their mutual pairwise interactions. Several types of SQ-MBD-based analyses are exemplified in real-world cases, including C70 fullerene surrounded by a large molecular ring, crambin protein, and selected systems from the S12L benchmark set. The manuscript is well written, the scientific presentation is sound, and the practical examples presented provide a clear demonstration of the usefulness of the approach. Overall, I am convinced that the submitted work represents a substantial contribution to the field of computer simulations of molecular and extended systems and deserves to be published in Nature Communications, provided the comments listed below are carefully addressed.*

Answer We thank the referee for the overall positive evaluation of our manuscript.

Point 1) *The original formulation of the MBD method, as presented in J. Chem. Phys. 138, 074106 (2013) allows a rather elegant and computationally inexpensive way the partitioning of MBD correlation energy into individual atomic contributions. As described in Sec. II of the above-mentioned work, the MBD energy can be expressed as*

$$E_{\text{MBD}} = -\frac{1}{2\pi} \int_0^\infty d\omega \sum_{n=2}^{\infty} \frac{1}{n} \text{Tr}\{(AT)^n\} \quad (2)$$

from which the following expression for the contribution of an atom to the energy can be deduced:

$$E_{\text{MBD},A} = -\frac{1}{2\pi} \int_0^\infty d\omega \sum_{n=2}^{\infty} \frac{1}{n} \text{Tr}\{(AT)^n\}_{A,A} \quad (3)$$

where $\{(AT)^n\}$ denotes a 3×3 block of the matrix (as defined in J. Chem. Phys. 138, 074106 (2013)). If I understand correctly, this piece of information should be identical to $(U_{\text{MBD}})_A$ (eq. 11 in SM). This alternative route to this quantity should be mentioned in the paper and, ideally, consistency of both approaches should be demonstrated.

Answer We sincerely thank the referee for having suggested this comparison among different methods for defining an atomic contribution to the MBD energy of a given system. The intra-fragment contribution to the MBD energy $(U_{\text{MBD}})_A$ does not coincide with $E_{\text{MBD},A}^{(\text{Ref})}$, as defined in Eq. (3) by the referee. Let us consider, for instance, an atomic partitioning of the system into fragments $A = 1, \dots, N_{\text{frag}}$. Then

$$E_{\text{MBD}} = \sum_{A=1}^{N_{\text{frag}}} (E_{\text{MBD}}^{\text{frag}})_A = \sum_{A=1}^{N_{\text{frag}}} \left[(U_{\text{MBD}})_A + \sum_{B \neq A} (V_{\text{MBD}})_{AB} \right] = \sum_{A=1}^{N_{\text{frag}}} (E_{\text{MBD}})_A . \quad (4)$$

As in general $\sum_B (V_{\text{MBD}})_{AB} < 0$ it follows that $(U_{\text{MBD}})_A \neq E_{\text{MBD},A}$. The observable that more directly compares with $E_{\text{MBD},A}^{\text{ACFD}}$ is the single fragment $(E_{\text{MBD}}^{\text{frag}})_A$ contribution to the total MBD energy, as their sums over all the fragment coincide. A detailed investigation of the information content of the two methods from both a theoretical and numerical point of view is beyond the scope of the present work.

However, a first numerical check of consistency between the two methods has been completed for the case of the C70@8PPA supramolecular complex and reported in the Supplementary Material. The results of our analysis confirm that the two reprojection methods do not coincide. We conjecture that the main difference between the two methods consists in the way the QDOs energy in the non-interacting ground state is distributed among the atoms. According to the reprojection procedure defined by the SQ-MBD method, the MBD energy of each 3D atomic QDO in the interacting system is calculated, assuming its ground state energy in the non-interacting system as reference. On the other hand, the ACFD-RPA reprojection method integrates the energy differences between the non-interacting and interacting system without explicitly referring each atomic QDO energy to its non-interacting ground state energy. Again, a detailed study of the inequivalence of these distinct reprojection methods in a more rigorous and systematic way is beyond the scope of the present work.

Actions A subsection has been added to the Supplementary Material reporting the results of the numerical comparison between the reprojection schemes based on SQ-MBD and ACFD-RPA.

Point 2) *The physical model behind MBD@rsSCS suffers from some serious issues not mentioned in the manuscript, which preclude a direct application of the method to some of the emblematic vdW systems, such as the transition metal dichalcogenides (see, e.g., Sec.*

2.3 of JCTC 12, 5920 (2016)). In the meantime, some of the authors of the submitted work introduced an improved version of MBD, called MBD-NL (PRL 124, 146401 (2020)), which is presumably free of such problems. Why did the authors choose to develop the SQ-MBD approach based on the MBD@rsSCS framework, if MBD-NL “increases the accuracy and efficiency of existing vdW functionals and is shown to be broadly applicable to molecules, soft and hard materials including ionic and metallic compounds, as well as interfaces between organic molecules and inorganic materials.”? This is very confusing because, to my understanding, the MBD- NL approach is meant to be the most accurate MBD-based approach available to date and as such it should eventually replace all the other MBD flavours. A comment on that would be useful. In any case, it would be fair if the problems of the MBD@rsSCS are mentioned in the paper

Answer We agree with the remark of the referee and his suggestion on how to improve the main text. We emphasize that the SQ-MBD formalism can be easily extended to the MBD-NL method without any particular modification. The SQ-MBD model is based on the integrable MBD Hamiltonian, which maintains the same form in both the MBD-NL and MBD@rsSCS methods. The only distinction between these two methods lies in the procedure employed to determine the parameters governing the atom-in-molecule response, namely the C_6 coefficients and polarizabilities. We expect qualitative differences between the more accurate MBD-NL method and MBD@rsSCS method, which is easier to implement. In future work, the SQ-MBD formulation will be integrated with the MBD-NL method, especially for applications to molecular systems where the use of MBD-NL is particularly appropriate, i.e., in ionic and hybrid metal-organic systems.

Actions We have included in the Methods section a few sentences discussing the limitations of MBD@rsSCS, and how the SQ-MBD method can be eventually extended to MBD-NL.

Point 3) *A minor point – I guess the last sign in the first line of eq.8 in SM should be “-” instead of “+”*

Answer We sincerely thank the referee for his very careful reading of the details of our manuscript, and for pointing our attention to this typographical error.

Actions The typo has been corrected.

REVIEWER COMMENTS

Reviewer #1 (Remarks to the Author):

In the revised version of the manuscript, the Authors have fully addressed the points raised in my previous report.

My suggestion is that this paper can be accepted for publication in Nature Communications.

Reviewer #2 (Remarks to the Author):

2nd review of Gori et al., NCOMMS-23-15403

This is a technically interesting extension of the MBD@rsSCS approach of Tkatchenko and coworkers that is now widely used as a dispersion model for DFT calculations, yet it continues to feel to me like a modest theoretical extension. The coarse-graining that the authors suggest is already available from the "first-quantized" version of the model, insofar as the MBD Hamiltonian has atomic indices therefore readily admits an atomic partition. Other extensions, e.g., to quantum information theory or properties calculations are only speculative at this stage. The collective nature of the MBD eigenmodes, which the authors describe here using plasmon language, has already been described by Tkatchenko in at least two high-profile publications, one that I mentioned in my previous review [Nat. Commun. 8, 14052 (2017), 10.1038/ncomms14052] and also PNAS 109, 14791 (2012), 10.1073/pnas.1208121109], the latter based on a slightly older version of the model but nevertheless the manner in which the model can be used to analyze long-range correlations is already clear in that much older work, and has been repeated elsewhere more recently to examine long-range correlations (arising from dispersion interactions) in solvated polypeptides, as described by the authors in their rebuttal letter. Viewed in this context, the change-of-basis transformation that is the primary result in the present work feels more appropriate for a specialized journal.

Reviewer #3 (Remarks to the Author):

The authors' answered to my points of criticism are acceptable and I can now recommend the submitted manuscript for publication.

Answer to Referees of NCOMMS-23-15403-Second Round:
Second Quantization Approach to Many-Body Dispersion

Interactions:

Insights for Chemical and Biological Systems

Matteo Gori, Philip Kurian, and Alexandre Tkatchenko

(Dated: Sep 2023)

REVIEWER #2

Statement *This is a technically interesting extension of the MBD@rsSCS approach of Tkatchenko and coworkers that is now widely used as a dispersion model for DFT calculations, yet it continues to feel to me like a modest theoretical extension. The coarse-graining that the authors suggest is already available from the “first-quantized” version of the model, insofar as the MBD Hamiltonian has atomic indices and therefore readily admits an atomic partition.*

Answer As we mentioned in our previous response, the transition from first quantization to second quantization in quantum many-body systems results in a disparity in the accessibility of information between the two approaches. In general, the SQ-MBD method offers a paradigm shift in the analysis of MBD effects, providing access to a complementary picture with respect to the one offered by a first-quantized MBD description.

In particular, the second-quantization approach, known as SQ-MBD, provides an enhanced capability to calculate the matrix elements of operators that are expressed as polynomials in the creation and annihilation operators. This enhancement significantly facilitates the computation of the novel quantities presented in our manuscript, ranging from fragment contributions to MBD energies to quantum information metrics among QDO subsets in the MBD ground state.

Statement *Other extensions, e.g., to quantum information theory or properties calculations, are only speculative at this stage.*

Answer In the section dedicated to mutual information, we present the computation of a quantum information observable, i.e. the quantum mutual information, and we present an extended analysis of the network of correlation among fragments due to QDO correlations in the MBD ground state. This application is a clear example of the application of quantum information techniques (the calculation of the quantum information entropy and the quantum mutual information for a subset of modes in a multimodal Gaussian state) to the analysis of MBD modes in a realistic biomolecular system.

Actions Taken We have extended and clarified the application of QI tools to analyze the ground state of the MBD Hamiltonian. Moreover, we have introduced a new analysis of the network of quantum correlations in the MBD ground state among QDOs belonging to

different residues in a biomolecule. This additional analysis will find applications in future studies of intra- and intermolecular allosteric pathways supported by quantum electronic density fluctuations.

Statement *The collective nature of the MBD eigenmodes, which the authors describe here using plasmon language, has already been described by Tkatchenko in at least two high-profile publications, one that I mentioned in my previous review [Nat. Commun. 8, 14052 (2017), 10.1038/ncomms14052] and also PNAS 109, 14791 (2012), 10.1073/pnas.1208121109], the latter based on a slightly older version of the model but nevertheless the manner in which the model can be used to analyze long-range correlations is already clear in that much older work, and has been repeated elsewhere more recently to examine long-range correlations (arising from dispersion interactions) in solvated polypeptides, as described by the authors in their rebuttal letter. Viewed in this context, the change-of-basis transformation that is the primary result in the present work feels more appropriate for a specialized journal.*

Answer The approach and results obtained within the SQ-MBD model and shown in our manuscript are original and provide altogether distinct insights compared to previous work on the established MBD model. In particular, the visualization of the MBD ground state, the projection of the MBD energy into fragment contributions, the calculation of MBD transition dipoles, and the quantum mutual information analysis presented in our current manuscript have not appeared in any prior work, including those mentioned by the referee.

For instance, our MBD energy projection method into fragment contributions is based on the MBD ground state of a given structure and offers completely different non-perturbative insights with respect to other methods, including the one presented in the paper by Di Stasio et al., *PNAS* **109**, 14791 (2012), which is based on perturbation theory.

Moreover, we have added an extensive discussion in the main text on the analysis of the bonding/antibonding character of the MBD modes in supramolecular dimer complexes, providing qualitatively different insights with respect to the analysis presented in J. Hermann et al., *Nat. Comm.* **8**, 14052 (2017), where the contributions to the MBD binding energies required the calculation of the MBD ground state for each isolated monomer. These results demonstrate how the SQ-MBD method has provided key access to a rich playground of collective charge density fluctuations, whose implications for chemistry and biology have been detailed in our manuscript.

Actions Taken A section has been incorporated into the main text, focusing on the examination of “bonding” and “antibonding” MBD modes within supramolecular dimers. Furthermore, in this same section, we have examined the distinctions between our approach and alternative methods for analyzing MBD modes in supramolecular systems, such as those adopted in the works mentioned by the referee.

Our study of MBD ground-state correlations in a protein based on quantum information tools has been deepened and extended, including the construction of a network of MBD correlations to identify the eigenvector centrality of each residue. We have also provided commentary on the significance of such a network-based representation of intramolecular MBD correlations, particularly in the context of their relevance in biomolecular information processing.

REVIEWERS' COMMENTS

Reviewer #3 (Remarks to the Author):

In my opinion, the paper was publishable already before this iteration. The discussion newly added in respond to Reviewer's 2 points is scientifically sound and nicely illustrated the utility of MBD formulated in the second quantisation formalism presented in this work. Altogether, I strongly support publication of this work in its present form.